# Structure and antagonism of the receptor complex mediated by human TSLP in allergy and asthma

Kenneth Verstraete[1,2], Frank Peelman[3], Harald Braun[1,4], Juan Lopez[5,6], Dries Van Rompaey[7], Ann Dansercoer[1,2], Isabel Vandenberghe[2], Kris Pauwels[8,9], Jan Tavernier[3], Bart N. Lambrecht[1,10], Hamida Hammad[1,10], Hans De Winter[7], Rudi Beyaert[1,4], Guy Lippens[5,11] & Savvas N. Savvides[1,2]

The pro-inflammatory cytokine thymic stromal lymphopoietin (TSLP) is pivotal to the pathophysiology of widespread allergic diseases mediated by type 2 helper T cell (Th2) responses, including asthma and atopic dermatitis. The emergence of human TSLP as a clinical target against asthma calls for maximally harnessing its therapeutic potential via structural and mechanistic considerations. Here we employ an integrative experimental approach focusing on productive and antagonized TSLP complexes and free cytokine. We reveal how cognate receptor TSLPR allosterically activates TSLP to potentiate the recruitment of the shared interleukin 7 receptor α-chain (IL-7Rα) by leveraging the flexibility, conformational heterogeneity and electrostatics of the cytokine. We further show that the monoclonal antibody Tezepelumab partly exploits these principles to neutralize TSLP activity. Finally, we introduce a fusion protein comprising a tandem of the TSLPR and IL-7Rα extracellular domains, which harnesses the mechanistic intricacies of the TSLP-driven receptor complex to manifest high antagonistic potency.

[1] VIB-UGent Center for Inflammation Research, Zwijnaarde, Ghent 9052, Belgium. [2] Laboratory for Protein Biochemistry and Biomolecular Engineering, Department of Biochemistry and Microbiology, Ghent University, Ghent 9000, Belgium. [3] VIB-UGent Center for Medical Biotechnology, Ghent 9000, Belgium. [4] Department of Biomedical Molecular Biology, Ghent University, Zwijnaarde, Ghent 9052, Belgium. [5] Unité de Glycobiologie Structurale et Fonctionnelle-CNRS UMR8576, Université de Lille, Villeneuve d'Ascq 59655, France. [6] Sciences Department-Chemistry, Pontifical Catholic University of Peru, Lima 32, Peru. [7] Laboratory of Medicinal Chemistry, Department of Pharmaceutical Sciences, University of Antwerp, Wilrijk 2610, Belgium. [8] VIB-VUB Center for Structural Biology, Brussels 1050, Belgium. [9] Structural Biology Brussels, Bio-Engineering Sciences Department, Vrije Universiteit Brussel, Brussels 1050, Belgium. [10] Department of Respiratory Medicine, Ghent University Hospital, Ghent 9000, Belgium. [11] LISBP, Université de Toulouse, CNRS, INRA, INSA, Toulouse 31400, France. Correspondence and requests for materials should be addressed to S.N.S. (email: savvas.savvides@ugent.be).

Thymic stromal lymphopoietin (TSLP)[1,2], is an interleukin-2 (IL-2) family cytokine produced in response to pathogenic stimuli by skin keratinocytes and epithelial cells in the lung and gut. It regulates immunity at barrier surfaces by driving the activation of immature dendritic cells (DCs), mast cells, basophils, eosinophils and lymphocytes into a type 2 polarizing phenotype[3,4]. TSLP initiates intracellular signalling by establishing a complex with its specific receptor, TSLPR (encoded by CRLF2) (refs 5,6) and IL-7Rα. Notably, the latter also serves together with the common gamma-chain (γc) receptor in signalling complexes driven by IL-7 to regulate T-cell development and homoeostasis[7].

The downside of aberrant signalling by TSLP has grave consequences for human health and imprints a massive healthcare and socioeconomic footprint. This is because type 2 helper T cell (Th2)-mediated inflammatory responses primed by activated DCs, are pivotal for the onset of widespread allergic diseases of the airways, skin and gut[8]. In fact, TSLP is now widely considered to underlie some of the most prevalent inflammatory allergic disorders, such as the atopic diseases (asthma, atopic dermatitis and atopic rhinitis), chronic obstructive pulmonary disease (COPD) and eosinophilic esophagitis[9–12], and has been annotated as a genetic risk factor for the development of asthma[13–15] and eosinophilic esophagitis[16]. Furthermore, a staggering 70% of atopic dermatitis clinical cases go on to develop asthma via the 'allergic march' (also known as 'atopic march')[17], and adult asthmatics are strongly predisposed for acquiring COPD (ref. 18). Several recent developments have expanded the pathophysiological profile of TSLP. First, TSLP was shown to provide a signalling link between the skin epithelium and neuronal cells to trigger itch associated with atopic dermatitis[19]. Second, TSLP was shown to contribute to the development of psoriasis, a widespread autoimmune disease, by regulating IL-23 production by DCs[20]. Third, TSLP may drive tumour progression in breast- and pancreatic cancer[21,22] but also manifest tumour protective effects[23–26], while genetic rearrangements and mutations in the TSLPR gene (CRLF2) are found in paediatric acute lymphoblastic leukaemia (ALL)[27]. However, the role of TSLP in cancer is controversial[8,28]. Fourth, TSLP was shown to upregulate IL-9 production in vivo to promote Th9 cell-induced allergic inflammation suggesting a possible interplay between the two cytokines and their hallmark Th2 and Th9 responses in allergy[29]. Finally, TSLP has been linked to neutrophil-mediated killing of bacteria trough interactions with the complement system[30].

Such a broad pathophysiology profile and the soaring rates of atopic and autoimmune diseases in the second half of the 20th century have motivated therapeutic targeting of TSLP and TSLP-mediated signalling[31,32]. For instance, blockade of TSLPR in a primate animal model was shown to attenuate allergic inflammation[33], and TSLP was shown to be pivotal for the development of resistance to corticosteroid treatment during airway inflammation[34]. More recently, the combinatorial ablation of TSLP, IL-25 and IL-33 has displayed therapeutic potential in mouse disease models of inflammation and fibrosis[35]. Notably, the validity of TSLP as a therapeutic target in humans was demonstrated in a clinical trial in which asthmatic patients were treated with an anti-TSLP monoclonal antibody[36].

In this study, we delineate the molecular, structural and mechanistic principles underpinning the extracellular assembly of the pro-inflammatory signalling complex driven by human TSLP and its antagonism by the therapeutic monoclonal antibody Tezepelumab (AMG-157/MEDI9929). We further describe the development of fusion proteins featuring tandem arrangements of the ectodomains of human TSLPR and IL-7Rα as potent antagonists of human TSLP signalling.

## Results

**Reconstitution and cooperativity of the TSLP complex.** Prior studies had suggested that the signalling complex mediated by human TSLP proceeds through an initial binary complex between TSLP and TSLPR to enable recruitment of IL-7Rα (refs 5,6,37). To determine the assembly order and kinetic profile underlying the TSLP:TSLPR:IL-7Rα complex we performed real time in vitro interaction studies via bio-layer interferometry (BLI) using mammalian-derived glycosylated TSLP, IL-7 and soluble TSLPR and IL-7Rα (Supplementary Fig. 1A). In accordance to prior observations human TSLP could only be produced in HEK293 cells upon abolishing its putative furin cleavage site[38]. Firstly, we determined that TSLPR binds to TSLP with high-affinity ($K_D = 32$ nM) and fast kinetics ($k_a = 1.7 \times 10^5$ M$^{-1}$s$^{-1}$ and $k_d = 5.2 \times 10^{-3}$ s$^{-1}$) (Fig. 1a). In contrast, IL-7Rα, which was able to bind to cognate IL-7 (Supplementary Fig. 1B), showed no apparent binding to TSLP alone (Fig. 1b)[39]. However, IL-7Rα associated with preformed TSLP:TSLPR binary complex with high-affinity ($K_D = 29$ nM; $k_a$ of $1.23 \times 10^5$ M$^{-1}$s$^{-1}$; $k_d$ of $3.6 \times 10^{-3}$ s$^{-1}$) (Fig. 1c). Thus, priming of human TSLP by its cognate receptor, TSLPR, is a mechanistic prerequisite for the recruitment of shared IL-7Rα to the extracellular ternary complex.

Such initial mechanistic insights formed the basis for a strategy to biochemically reconstitute the TSLP:TSLPR:IL-7Rα complex for structural studies. To facilitate the growth of well-diffracting crystals towards structural characterization of the complex at high-resolution by X-ray crystallography, we focused on the production of minimally glycosylated ternary complexes. We thus produced non-glycosylated bioactive human TSLP lacking a basic cassette ($^{127}$RRKRK$^{131}$) (ref. 38) and the IL-7Rα ectodomain via in vitro refolding from inclusion bodies produced in E. coli[37,40]. In parallel, we were able to produce N-glycosylation variants of TSLPR (TSLPR$^{N47Q}$, TSLPR$^{N47Q/N101Q}$ and TSLPR$^{N47Q/N169Q}$) in HEK293S-TetR MGAT1$^{-/-}$ cells[41,42]. Following enzymatic trimming of residual TSLPR glycosylation, ternary TSLP:TSLPR:IL-7Rα complexes were assembled and isolated in a sequential manner by size-exclusion chromatography (SEC) (Fig. 1d,e), and were found to be highly homogeneous (Fig. 1f) and to adopt monodisperse assemblies obeying 1:1:1 stoichiometry as characterized by coupling SEC to multi-angle laser light scattering (MALLS) (Fig. 1g). Crystallization trials using purified TSLP$^{\Delta127–131}$:TSLPR$^{N47Q}$:IL-7Rα complex lead to optimized crystals that diffracted synchrotron X-rays to 2.55 Å resolution, and enabled determination of the crystal structure of the human TSLP:TSLPR:IL-7Rα complex by molecular replacement (Fig. 1h, Table 1).

**TSLP evokes receptor–receptor interactions.** Our crystallographic analysis contributes structural insights at high-resolution of human TSLP and TSLPR (Supplementary Fig. 1C) and reveals how TSLP wedges between TSLPR and IL-7Rα to mediate a T-shaped extracellular assembly (Fig. 2a), as further supported by small-angle X-ray scattering (Supplementary Fig. 2; Supplementary Table 1). TSLP employs two opposing surface patches to interact with the elbow tips of the cytokine-binding homology regions (CHRs) of TSLPR (site I) and IL-7Rα (site II), which allow the membrane-proximal parts of the two receptors to engage in heterotypic receptor–receptor interactions (site III) (Fig. 2a). TSLP and TSLPR display pronounced electrostatic complementarity spanning the entire site I, with TSLP presenting a positively charged surface patch associating with the negatively charged interdomain elbow of TSLPR (Fig. 2b). This suggests that long-range electrostatic interactions may play an important role in attracting TSLP to TSLPR at the cell surface to establish the mechanistically critical binary complex. Interestingly, electrostatic

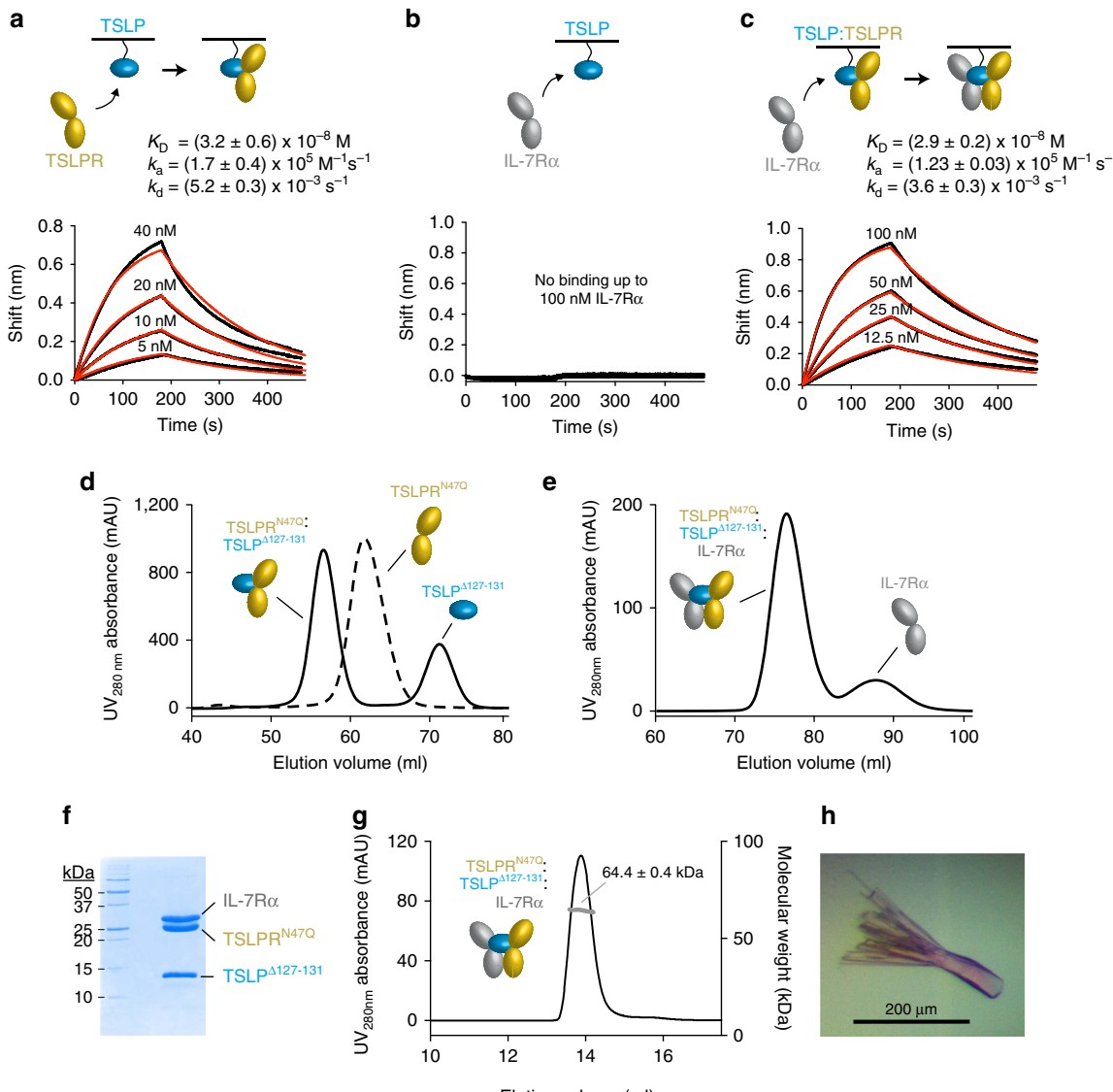

**Figure 1 | The TSLP signalling complex assembles via a cooperative stepwise mechanism.** (**a–c**) BLI data traces (black) and fitted 1:1 binding model (red) are plotted as the spectral nanometre shift in function of time for the interaction of TSLPR with TSLP (**a**), IL-7Rα with TSLP (**b**) and IL-7Rα with preformed TSLP:TSLPR complex (**c**). 4 nm of biotinylated TSLP was loaded for experiments (**a–c**). In experiment (**c**) TSLP-coated biosensors were incubated with 320 nM TSLPR and 320 nM of TSLPR was included in kinetics buffer and IL-7Rα samples. The reported $K_D$-, $k_a$- and $k_d$-values represent average values and their s.d. from three technical replicate experiments. (**d**) SEC elution profiles on a Superdex 75 16/600 column of EndoH-treated TSLPR$^{N47Q}$ (dashed line) and TSLP$^{\Delta127-131}$:TSLPR$^{N47Q}$ complex (solid line) plotted as the ultraviolet absorbance at 280 nm in function of elution volume. (**e**) SEC elution profile on a Superdex 200 16/600 column of TSLP$^{\Delta127-131}$:TSLPR$^{N47Q}$:IL-7Rα complex plotted as the ultraviolet absorbance at 280 nm in function of elution volume. (**f**) Coomassie-stained reducing SDS–PAGE gel of glycan-minimized ternary complex used for crystallization trials. The theoretical protein molecular weights for IL-7Rα, TSLPR and TSLP are 25.7, 24.0 and 14.6 kDa. (**g**) SEC elution profile on a Superdex-200 10/300 GL column of glycan-minimized ternary TSLP complex plotted as the ultraviolet absorbance at 280 nm (left $Y$-axis) as a function of elution volume. The molecular weight (right $Y$-axis) as determined by MALLS is reported as the number average molecular mass and its s.d. (**h**) Crystals of the TSLP$^{\Delta127-131}$:TSLPR$^{N47Q}$:IL-7Rα complex.

potential calculations on the TSLP:TSLPR binary complex show that this molecular entity would project a negative electrostatic potential, which would render it compatible with the positive electrostatic potential of unbound IL-7Rα.

Consistent with annotations of human TSLP as a member of the IL-2 family of cytokines, its mature sequence (residues 29–159) adopts a four-helix bundle with 'up-up-down-down' topology stabilized by three disulfide bridges (Cys34-Cys110, Cys69-Cys75 and Cys90-Cys137), in which the four α-helices— designated αA, αB, αC, αD—are threaded via a *BC* loop and two long overhand *AB* and *CD* loop regions, with the latter largely invisible in the electron density maps (Fig. 2a; Supplementary

Fig. 3A). The functional role of the flexible *CD* loop containing the seven residue basic cassette (residues 125–131) remains enigmatic (Supplementary Fig. 3A). It has been hypothesized that its embedded furin cleavage site is linked to a mechanism limiting the availability of proinflammatory TSLP *in vivo*[43]. Moreover, it was recently shown that in nasal polyp tissues this loop region can be proteolytically processed to yield a biologically active nicked form of human TSLP (ref. 44). In addition, the positive charge density may mediate interactions with glycosoaminoglycans in the extracellular matrix, as proposed for IL-7 (ref. 45).

The structure of human TSLP is unique among helix bundle cytokines in three main ways. First, it adopts a rather open helix

**Table 1 | X-ray data set and refinement statistics\*,†.**

| | TSLP:TSLPR:IL-7Rα complex | TSLP:Fab complex |
|---|---|---|
| *Data collection* | | |
| Source | Proxima 2A (SOLEIL, France) | Proxima 2A (SOLEIL, France) |
| Detector | ADSC QUANTUM 315r | EIGER 9M |
| Wavelength (Å) | 0.98 | 0.98 |
| Space group | $C\,2$ | $P\,3_2\,1\,2$ |
| Cell dimensions | | |
| $a, b, c$ (Å) | 135.8, 66.6, 92.0 | 51.7, 51.7, 370.0 |
| $\alpha, \beta, \gamma$ (°) | 90.0, 109.2, 90.0 | 90.0, 90.0, 120.0 |
| Resolution (Å) | 50.0–2.56 (2.72–2.56) | 55.0–2.30 (2.44–2.30) |
| ‡Wilson B (Å$^2$) | 69.9 | 47.33 |
| Completeness (%) | 97.8 (94.2) | 97.0 (83.8) |
| Redundancy | 3.2 (3.1) | 8.4 (4.2) |
| Mean $I/\sigma(I)$ | 14.2 (1.7) | 14.23 (1.6) |
| $R_{meas}$ (%) | 6.2 (77.6) | 11.0 (74.7) |
| $CC_{1/2}$ (%) | 99.8 (78.2) | 99.8 (64.2) |
| †*Refinement* | | |
| Resolution (Å) | 45.76–2.56 (2.67–2.56) | 44.80–2.30 (2.39–2.30) |
| No. reflections | 24,638 (2,738) | 25,275 (2,306) |
| $R_{work}/R_{free}$ | 0.1912/0.2176 (0.2674/0.3351) | 0.1891/0.2150 (0.3078/0.3914) |
| No. non-H atoms | 4,033 | 4,120 |
| Protein | 3,966 | 3,997 |
| Ligands | 38 | 15 |
| Water | 29 | 108 |
| Average ADP (Å$^2$) | 89.40 | 51.90 |
| Protein | 89.30 | 52.00 |
| Ligands | 118.30 | 67.60 |
| Water | 69.70 | 48.50 |
| r.m.s.d.'s | | |
| Bond lengths (Å) | 0.014 | 0.003 |
| Bond angles (°) | 1.71 | 0.77 |

\*Values in parentheses correspond to the highest-resolution shell.
†Final refinement was performed in autoBuster 2.10.2 for the TSLP:TSLPR:IL-7Rα complex and in PHENIX 1.9-1692 for the TSLP:Fab complex.
‡Maximum likelihood estimate of the Wilson B-factor from phenix.xtriage.

bundle core that is perforated by an elongated internal void volume ($\sim 120\,\text{Å}^3$) running from the αA-αC face to the αB-αC face of the helical bundle (Fig. 2c). Second, it harbours a fully buried structural water at the heart of the helical bundle, coordinated by a conserved trio of amino acids (Trp148, Thr102, Thr83) (Fig. 2c; Supplementary Fig. 3A) suggesting that this central water molecule is an integral part of the protein fold. This notion is supported by molecular dynamics (MD) simulations of TSLP using an explicit solvent model, whereby TSLP devoid of this core water molecule rapidly acquires a new water molecule from solvent through a water channel between helices B and C (Supplementary Fig. 3B). Third, the conspicuously kinked αA in TSLP, a structural feature shared with IL-7 despite the lack of appreciable levels of sequence identity (Fig. 2d, Supplementary Fig. 3C), is hallmarked by a π-helical turn. Interestingly, MD simulations showed that in about 20% of the simulated frames a water molecule inserted into the π-helical turn of TSLP, seemingly to compensate for the interrupted hydrogen-bonding pattern of the main chain (Supplementary Fig. 3D), reminiscent of water-mediated stabilization of π-helical turns in diverse proteins[46].

**The *AB* loop in TSLP relays IL-7Rα recruitment.** The atypical open helical bundle core of TSLP and the intriguing π-helical turn in helix αA of TSLP prompted us to hypothesize that the priming of TSLP by TSLPR for recruitment of IL-7Rα might be linked to the intrinsic plasticity and dynamics of TSLP. To this end, we performed a series of nuclear magnetic resonance (NMR) experiments on isotopically labelled TSLP$^{\Delta127-131}$ and pursued

complementary MD simulations. Assignment of the NMR spectra by triple resonance spectroscopy on isotopically labelled TSLP$^{\Delta127-131}$ revealed that unbound TSLP comprises the four α-helices as delineated in the structure of TSLP bound to its receptors (Fig. 2e). Furthermore, $^1$H–$^{15}$N heteronuclear NOE analysis showed decreased NOE values for the overhand *AB* and *CD* loops, as well as for the N- and C-termini, reflecting the relative higher flexibility of these regions compared to the helical parts of the structure[47] (Fig. 2f).

Altogether with the structure of receptor-bound TSLP, these findings provide the rationale for tracing possible structural transitions in TSLP upon complex formation. In particular, TSLP employs the C-terminal half of αD (residues 142–152), the C-terminal short tail extending from αD (residues 153–158) and a continuous stretch of 10 residues located in the long overhand *AB*-loop region (residues 60–69) to interact with a complementary interaction epitope formed at the elbow tip of the CHR-module of TSLPR (Fig. 2a; Supplementary Table 2). On the basis of our NMR studies, the AB-loop and C-terminal tail would undergo significant conformational changes to achieve their observed bound state. This is additionally supported by extensive MD simulations for TSLP and TSLP:TSLPR (Fig. 2g,h), and might have profound mechanistic implications. This is because the *AB* loop provides a physical link to αA, which in turn is central to defining site II and the interactions of TSLP with IL-7Rα (Fig. 2a). Thus, our findings point to the possible role of the *AB* loop as a structural liaison between the two receptor binding sites on TSLPR (sites I and II) poised to relay a binding event to TSLPR at site I to prime TSLP for the cooperative recruitment of IL-7Rα at site II.

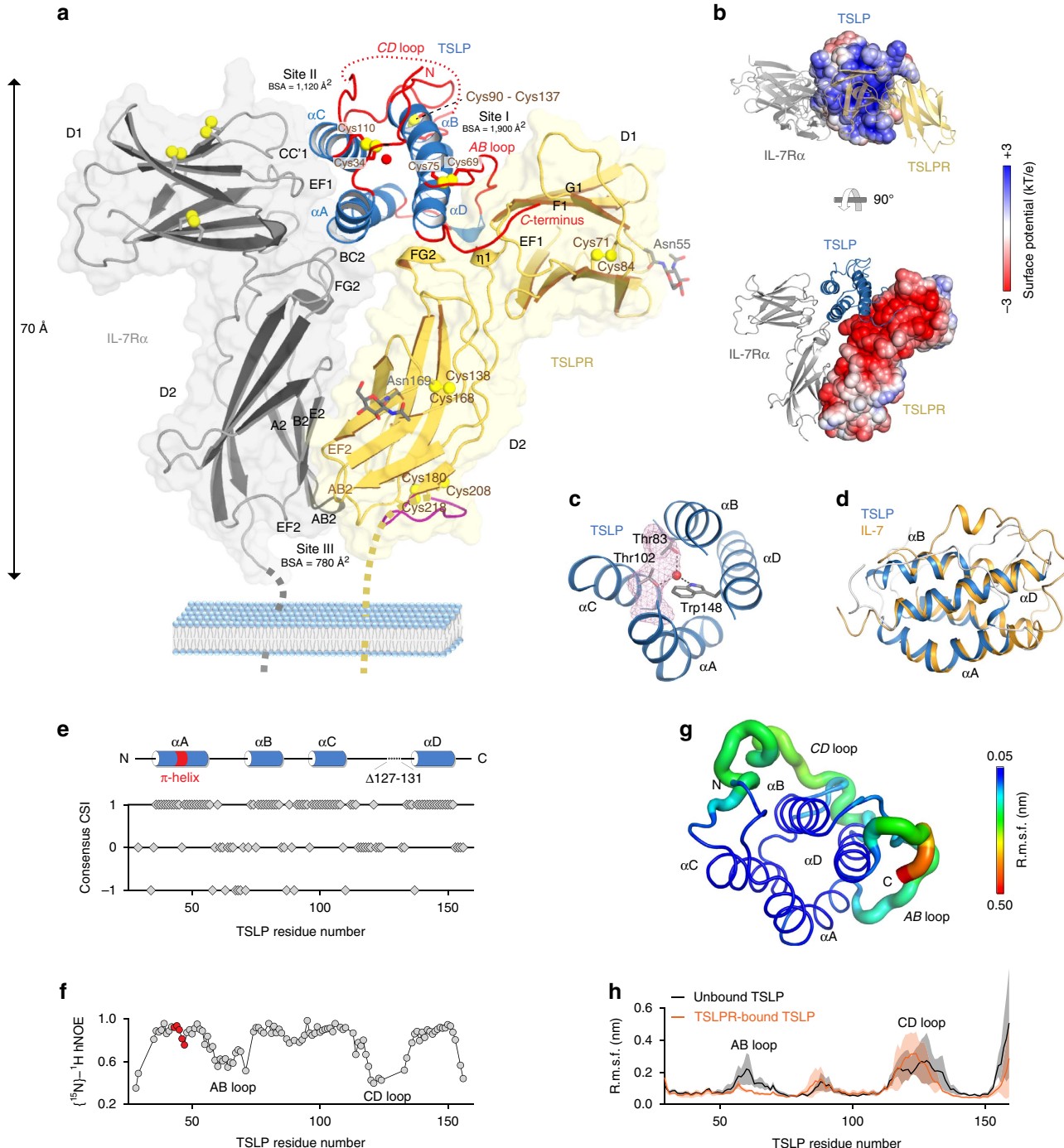

**Figure 2 | Structure of the TSLP:TSLPR:IL-7Rα complex and TSLP structural dynamics.** (**a**) View of the determined X-ray structure for the TSLP:TSLPR:IL-7Rα complex. TSLP (blue) is shown in cartoon representation with its four helices labelled as αA to αD. TSLP loop regions are highlighted in red with the disordered CD loop region represented as a dashed line. The extracellular regions of TSLPR (yellow) and IL-7Rα (grey), each comprising two FnIII-like domains, D1 and D2, are shown as cartoons overlaid onto transparent surface representations. TSLPR and IL-7Rα strand and loop regions contributing to sites I, II and III are labelled. Disulfide bridges are shown as yellow spheres. The water molecule in the core of TSLP is shown as a red sphere. Modelled GlcNAc moieties on TSLPR are shown as grey sticks. The C-terminal loop extending from TSLPR strand G2 is coloured purple. BSA: buried surface area. (**b**) Bottom and side view cartoon representations of the TSLP complex with the electrostatic potential at the solvent accessible surfaces shown for TSLP (top) and TSLPR (bottom). (**c**) The coordinated water molecule in the bundle core (red sphere) is located adjacent to an elongated internal void volume (pink mesh). (**d**) Structural comparison of human TSLP (blue) as seen in the TSLP:TSLPR:IL-7Rα complex and human IL-7 (orange) as seen in the IL-7:IL-7Rα complex (pdb 3DI2, chain A). (**e**) $^{13}$C-consensus chemical shift index (CSI) analysis of TSLP$^{Δ127-131}$. The location of crystallographically observed α-helices in the TSLP:TSLPR:IL-7Rα complex as annotated by DSSP is schematically shown on top with the π-helical turn coloured in red. (**f**) Backbone $^{15}$N-$^{1}$H heteronuclear NOE values measured on f TSLP$^{Δ127-131}$ plotted as a function of TSLP residue number. Data points for residues in the π-helical turn are coloured red. (**g**) Snapshot for one of five different 250 ns MD simulations runs for full-length TSLP is shown as a cartoon. Residues are coloured according to their average root mean square fluctuations (r.m.s.f.) of backbone positions over the five different MD runs. The width of the cartoon loop radius varies with the r.m.s.f. value. (**h**) Average backbone r.m.s.f. values and s.d. (shaded region) for unbound TSLP (black) and for TSLP in complex with TSLPR (red) plotted in function of TSLP residue number. Average r.m.s.f. values and s.d. were calculated from five 250 ns MD-runs, each with a different starting model for TSLP.

Our NMR analysis show that among all residues defining the four α-helices of TSLP, Thr46 and Ile47 in the π-helical turn midway αA exhibit the highest degree of flexibility in unbound TSLP (Fig. 2f), yet they become well-ordered at the TSLP:IL-7Rα interface. We therefore wondered about the origin of the structural features of the atypical αA of TSLP and about its possible role in IL-7Rα recruitment. Indeed, analysis of the NOE strips through each of the resonances of residues in αA showed that the relative intensity of the amide-amide proton cross peak compared to the diagonal peak is 15–25% for all amide proton pairs, except for the Thr46 $H_N$—Ile47 $H_N$ NOE where the normalized cross peak rises to 54% (Supplementary Fig. 3E). This discrepancy agrees with the distances between consecutive amide protons in αA in the bound state of TSLP, which measure at the expected $2.7 \pm 0.1$ Å throughout αA, except for the helical kink region, where the Thr46 $H_N$—Ile47 $H_N$ distance shortens to 2.1 Å, while the adjacent Ile47 $H_N$-Ser48 $H_N$ distance increases to 2.9 Å. Thus, unbound TSLP in solution also displays the kinked αA and associated π-helical turn character before engaging to TSLPR and IL-7Rα. The mechanistic ramifications of this deduction are large. In the context of a TSLP helical bundle core that is not densely packed, the TSLP:TSLPR binary complex might facilitate positioning of αA relayed by the AB loop and the tethering of αA via Leu44 to TSLPR. This can be expected to provide an entropic advantage for recruiting IL-7Rα to the ternary complex.

**TSLP and TSLPR interact via an extensive polar interface**. The extracellular domain of TSLPR (residues 25–230) carries a single CHR module composed of two tandem fibronectin type III (FnIII)-domains, D1 and D2 (Fig. 2a). The membrane-distal $TSLPR_{D1}$ domain is characterized by a ABED/C′CFG topology stabilized by a Cys71-Cys84 disulfide bridge, while the membrane-proximal $TSLPR_{D2}$ domain displays a ABE/C′CFG topology with two disulfide bridges Cys138-Cys168 and Cys180-Cys218 (Fig. 2a). The latter tethers Cys218 in the loop region extending from strand G2 in $TSLPR_{D2}$ towards the transmembrane helix of TSLPR with Cys180 located in strand F2 (Supplementary Fig. 4). Intriguingly, TSLPR carries a solvent-accessible cysteine residue at position 208 in close proximity to disulfide bridge Cys180-Cys218. Although the role of this unusual triangle of cysteine residues is currently unknown, and while surface-exposed cysteines are rarely observed in receptor ectodomains, disulfide-mediated linkage of TSLPR and IL-7Rα in the context of somatic mutations in the juxtamembrane and transmembrane domains has been connected to the pathophysiology of B-ALL and T-ALL (refs 48,49).

Human TSLPR uses three regions to grasp TSLP and buries $\sim 1,900$ Å$^2$ of solvent exposed surface area: (1) the intersheet EF1 loop and N-terminal residues of the F strand of $TSLPR_{D1}$ (residues 91–96), (2) the C-terminal residues of $TSLPR_{D1}$ strand G and the interdomain linker connecting $TSLPR_{D1}$ and $TSLPR_{D2}$ (residues 110–116) and (3) the α-helical turn located in the FG2 loop of $TSLPR_{D2}$ (residues 192–195), (Fig. 2a; Supplementary Table 2). Altogether, the human TSLP cytokine-receptor interface has a pronounced polar footprint and allows us to trace the species-specificity of the TSLP:TSLPR interaction[37] (Supplementary Figs 3A and 5), providing a potentially key resource towards interrogating human TSLP activity in mouse models.

We subsequently leveraged such detailed structural information to identify functional hotspots at the TSLP:TSLPR interface (Fig. 3a) via cellular studies in vitro including a STAT5 activation assay (Supplementary Table 3). Even though human TSLP has been linked to a number of JAK-STAT signalling pathways, STAT5 activation by TSLP has emerged as a signalling

prerequisite for Th2 responses mediated by TSLP (refs 4,50). In a competition-based cellular TSLPR-binding assay employing wild type TSLP fused to secreted alkaline phosphatase (TSLP-SEAP) we first identified a set of TSLP mutants that failed to displace TSLP-SEAP from TSLPR (Fig. 3b). These TSLP mutants probed the importance of the triplet of arginine residues near the C-terminal region of TSLP at the TSLP:TSLPR interface (Arg149, Arg150 and Arg153) (Fig. 3a). In TSLP-induced STAT5 activity assays these TSLP mutants were still able to induce STAT5 activation, albeit with half maximal effective concentration ($EC_{50}$) values lowered by 1–3 orders of magnitude as compared with WT ($EC_{50} = 0.15$ pM). The TSLP-Arg149Ser/Arg150Ser double mutant had the most pronounced effect ($EC_{50} = 100$ pM) (Fig. 3c). To probe the importance of TSLPR site I interface residues we performed STAT5-based cellular activity assays with a set of TSLPR variants (Supplementary Table 3). Here, TSLPR-Asp92Ala, TSLPR-Trp112Ala and TSLPR-Trp112Arg displayed a greater than 1,000-fold reduced $EC_{50}$-value as compared to wild type (Fig. 3d). The apparent essential roles of TSLPR-Asp92 and TSLPR-Trp112 in TSLP recruitment are borne by our structural observations. TSLPR-Asp92 pairs via a bifurcated hydrogen-bond with TSLP-Arg153 and TSLP-Arg150, while Trp112 packs between αD and the AB loop of TSLP (Fig. 3a).

**IL-7Rα binds TSLP via a degenerate hydrophobic interface**. IL-7Rα performs a dual role at the cell surface: it constitutes the high-affinity receptor for IL-7 signalling, and functions as the co-receptor for TSLP-mediated signalling (Supplementary Fig. 6A). Our structural studies reveal that IL-7Rα employs a hydrophobic platform contributed by several residues in IL-7Rα$_{D1}$ (Val78, Leu100, Ile102) and IL-7Rα$_{D2}$ (Tyr159, Tyr212, Phe213) to clamp onto the AC-face of the TSLP helical bundle as defined by residues protruding from αA of TSLP (Ala41, Ala42, Leu44, Ser45, Thr46, Leu47, Lys49) and αC (Met97, Met100, Lys101, Ala104, Ala105, Ile108 and Trp109) (Figs 2a and 3e; Supplementary Table 2). With $\sim 1,120$ Å$^2$ of buried solvent accessible surface area, the hydrophobic TSLP:IL-7Rα interface (site II) is markedly more limited than the TSLP:TSLPR interface (site I).

To interrogate the importance of TSLP residues at the TSLP:IL-7Rα interface, we evaluated a set of TSLP variants carrying mutations at site II (Supplementary Table 3). While our selected set of single site TSLP mutants had no apparent effect, we found that a double TSLP mutant carrying Ser45Arg/Thr46Arg mutations at the π-helical turn of αA (Fig. 3f) showed reduced capacity in inducing TSLPR/IL-7Rα-mediated STAT5-signalling ($EC_{50} = 5.3$ pM versus for $IC_{50}$,WT $= 0.11$ pM), while the affinity towards TSLPR remained largely unaffected ($IC_{50} = 720$ pM versus $IC_{50}$,WT $= 320$ pM) (Fig. 3g). At site II, mutations in the hydrophobic EF loop region of IL-7Rα$_{D1}$ (Leu100Ser/Ile102Ser, $EC_{50} = 470$ pM) also led to a decreased signalling potential ($EC_{50} = 85$ pM) (Fig. 3d).

Comparisons with the human IL-7:IL-7Rα binary complex[51] show that IL-7Rα adopts a highly similar structure in the two complexes (r.m.s.d. $= 0.66$ Å for 195 Cα atoms) and offers preformed binding sites to either cytokine as evidenced by the structure of IL-7Rα in the absence of cytokine (Supplementary Fig. 6B). Although IL-7 and TSLP exhibit marginal sequence similarity (Supplementary Fig. 3C), IL-7Rα employs a near identical set of residues to interact with IL-7 and TSLP, burying 740 Å$^2$ and 630 Å$^2$ of solvent-accessible surface, respectively (Supplementary Fig. 6C,D). Thus, the cytokine binding degeneracy of IL-7Rα originates from a promiscuous hydrophobic platform at the IL-7Rα elbow tip combined with unique structural features shared between the TSLP and IL-7 cytokines.

**Receptor–receptor interactions potentiate TSLP signalling.** One of the observed hallmarks of the receptor complex mediated by human TSLP concerns the compact network of interactions between the membrane-proximal regions of $TSLPR_{D2}$ and IL-$7R\alpha_{D2}$ (Figs 2a and 4a; Supplementary Table 2). The ensuing

heterotypic receptor interface buries $\sim 780\ \text{Å}^2$ of solvent-accessible surface area contributed by the AB2, CC'2 and EF2 loops of $TSLPR_{D2}$, and the ABE2-face, and AB2 and EF2 loops of IL-$7R\alpha_{D2}$. The interface displays several electrostatic interactions and close van der Waals contacts, such as the packing of TSLPR-

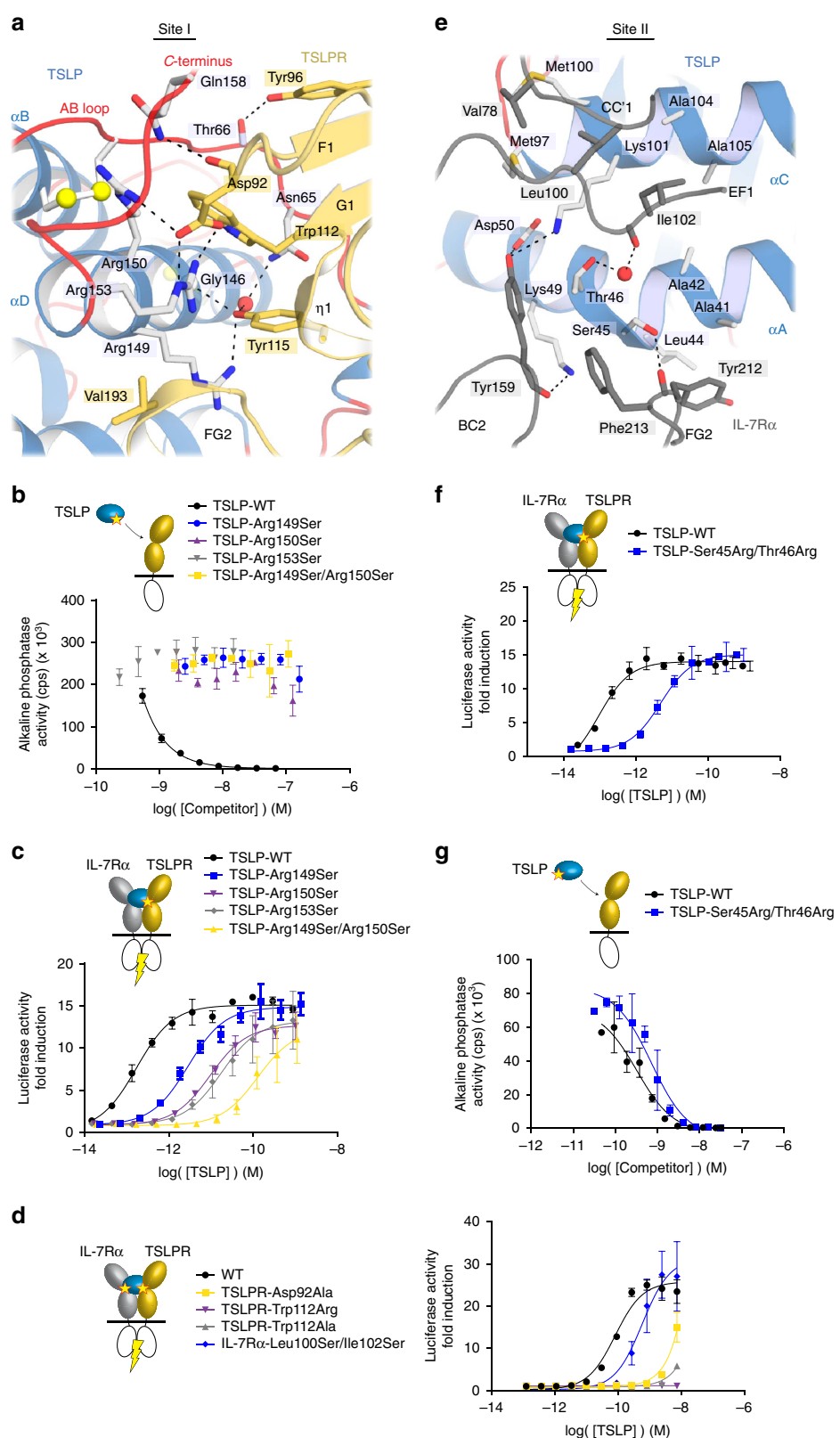

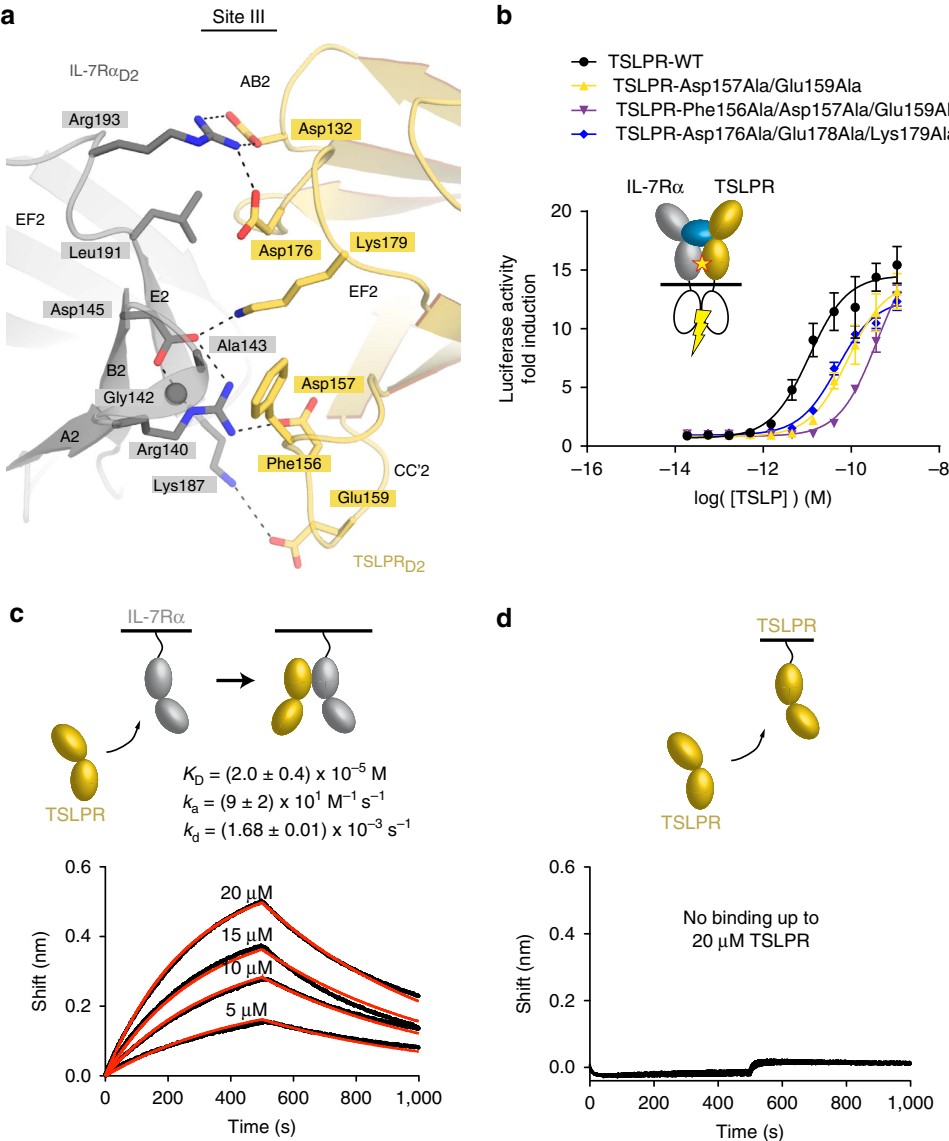

**Figure 4 | Membrane-proximal receptor–receptor interactions potentiate TSLP signalling.** (**a**) Detail of the IL-7Rα:TSLPR interface (site III) as viewed from the membrane-proximal side. Interface residues are shown as sticks. Hydrogen bonds and salt bridges are indicated with a dashed line. (**b**) TSLP-induced STAT5 activity assay in cells expressing WT or mutant forms of TSLPR. The EC$_{50}$-values were 11 pM for the control and 85 pM for TSLPR-Asp157Ala/Glu159Ala, 400 pM for TSLPR-Phe156Ala/Asp157Ala/Glu159Ala and 49 pM TSLPR-Asp176Ala/Glu178Ala/Lys179Ala. Each experiment was carried out in triplicate; data shown are averages, and error bars were calculated as s.e.m. (**c,d**) BLI data traces (black) and the fitted 1:1 binding model (red) are plotted as the spectral nanometre shift in function of time for the interaction of soluble TSLPR with immobilized IL-7Rα (**c**) or TSLPR (**d**). The reported averaged $K_D$-, $k_a$- and $k_d$-values and their s.d. are derived from three technical replicate experiments. Biosensing surfaces were generated by loading 2.5 nm of biotinylated IL-7Rα for experiment (**c**) and 1.5 nm of biotinylated TSLPR for experiment (**d**). The concentration range of soluble TSLPR in **d** was identical as in **c**.

**Figure 3 | Structure-based dissection of TSLP cytokine-receptor interfaces.** (**a**) Detail of the TSLP:TSLPR interface (site I). Interface residues are shown as sticks. Water molecules are shown as red spheres. Hydrogen bonds and salt bridges are indicated with a dashed line. (**b**) Competitive binding assay measuring displacement by either wild type (WT) or mutated TSLP of bound TSLP-SEAP fusion protein from HEK293T cells expressing TSLPR. (**c**) STAT5 activity induced by WT or mutated TSLP, as measured using a luciferase-based reporter system in HEK293T cells expressing WT TSLPR and IL-7Rα. The EC$_{50}$-values were 0.15 pM for TSLP-WT, 2.8 pM for TSLP-Arg149Ser, 9.5 pM for TSLP-Arg150Ser, 100 pM for TSLP-Arg149Ser/Arg150Ser and 19 pM for TSLP-Arg153Ser. (**d**) TSLP-induced STAT5 activity assay in cells expressing WT or mutant forms of either TSLPR or IL-7Rα. The EC$_{50}$-values were 85 pM for the control, 87 nM for TSLPR-Asp92Ala and 470 pM for IL-7Rα-Leu100Ser/Ile102Ser. For mutants TSLPR-Trp112Ala and TSLPR-Trp112Arg the EC$_{50}$-value could not be accurately determined. (**e**) Detail of the TSLP:IL-7Rα interface (site II). Interface residues are shown as sticks. Water molecules are shown as red spheres. Hydrogen bonds and salt bridges are indicated with a dashed line. (**f**) STAT5 activity induced by WT or TSLP-Ser45Arg/Thr46Arg in cells expressing wild-type TSLPR and IL-7Rα. The EC$_{50}$ values were 0.11 pM for TSLP-WT and 5.3 pM for TSLP-Ser45Arg/Thr46Arg. (**g**) Competitive binding assay measuring displacement by either WT (IC$_{50}$ = 320 pM) or TSLP-Ser45Arg/Thr46Arg (EC$_{50}$ = 720 pM) from bound TSLP-SEAP fusion protein from HEK293T cells expressing TSLPR. Each experiment was carried out in triplicate; data shown are averages, and error bars were calculated as s.e.m.

Phe156 against a hydrophobic patch defined by Gly142 and Ala143 in the AB2-loop of IL-7R$\alpha_{D2}$ at the base of the interface (Fig. 4b).

To evaluate the importance of the heterotypic TSLPR:IL-7R$\alpha$ interactions for TSLP-induced signalling we evaluated a set of TSLPR site III mutants via our STAT5-activation assays (Fig. 4b). While single-site site III mutants of TSLPR (Supplementary Table 3) had no apparent effect, combinations of mutations in the CC'2 loop (Asp157Ala/Glu159Ala and Phe156Ala/Asp157Ala/Glu159Ala) or EF2 loop (Asp176Ala/Glu178Ala/Lys179Ala) of TSLPR$_{D2}$ (Fig. 4a) showed decreased STAT5 activity as compared to wild type TSLPR. These results indicate that TSLPR$_{D2}$:IL-7R$\alpha_{D2}$ interactions are required for efficient activation of TSLP-induced signalling. We next probed the potential interaction between the receptor ectodomains in the absence of TSLP and measured a low, albeit appreciable, affinity ($K_D = 20$ μM), contrasting the lack of any measurable interaction between the TSLPR ectodomains (Fig. 4c,d). This suggests that TSLPR and IL-7R$\alpha$ are predisposed to interact under certain conditions at the cell membrane and might provide the basis for rationalizing the mechanism of disease-related mutations localizing in the membrane-proximal regions of the two receptors[52].

**Receptor fusion proteins are potent TSLP antagonists.** TSLP is increasingly gaining a central role in the pathophysiology of allergic diseases. To identify novel TSLP antagonists, we designed TSLP cytokine traps[53] by fusing the TSLPR and IL-7R$\alpha$ extracellular regions in both orientations with a flexible (Gly-Gly-Ser)$_{20}$-linker, hereafter termed TSLP-trap1 and TSLP-trap2 (Fig. 5a), and produced them in stably transfected T-Rex-293 cells (Fig. 5b,c; Supplementary Fig. 7A). We found that TSLP-trap1 binds 250-fold stronger to TSLP ($K_D = 120$ pM) (Fig. 5d) than the unlinked receptor ectodomains do (Fig. 1c). Importantly, the corresponding kinetic profile shows that such high-affinity can be traced to a drastically reduced dissociation rate constant ($k_d = 2 \times 10^{-5}$ s$^{-1}$) as compared to the dissociation rate of the unfused ectodomains (Fig. 1c). A similar binding profile was observed for TSLP-trap2 (data not shown).

To compare the binding properties of our TSLP-traps to Tezepelumab (AMG-157/MEDI9929) (ref. 54), to our knowledge the most potent anti-TSLP antagonist to date, we produced AMG-157 and its Fab fragment in HEK293T cells (Supplementary Fig. 8A,B). AMG-157 is a fully human neutralizing immunoglobulin G subclass 2 (IgG2) anti-TSLP monoclonal antibody (mAb) and was recently shown to alleviate most measures of both early and late asthmatic responses in patients with mild allergic asthma[36]. AMG-157$_{Fab}$ displays comparable affinity and binding kinetics to TSLP when compared to TSLP-trap1 (Fig. 5e). However, cellular activity assays in HEK293T cells at 10 pM TSLP (EC$_{50}$ = 9 pM) demonstrated that TSLP-trap1 and TSLP-trap2 are 20–30 fold more potent in inhibiting TSLP-induced STAT5 signalling (IC$_{50}$ = 67 and 44 pM, respectively) (Fig. 5f, Supplementary Fig. 7B) compared to AMG-157 mAb and its Fab fragment (IC$_{50}$ = 1.4 and 1.7 nM, respectively) (Fig. 5f). Remarkably, the TSLP-traps show about 1,000-fold higher inhibitory potency over equimolar mixtures of unlinked TSLPR and IL-7R$\alpha$ (IC$_{50}$ = 49 nM) (Fig. 5g), suggesting that fusion of the two receptor ectodomains harnesses certain key mechanistic features underlying the TSLP-receptor complex. Parallel assays at 100 pM TSLP resulted in analogous observations but with overall higher IC$_{50}$-values (Supplementary Fig. 7C). Finally, neither TSLP-trap1 nor TSLP-trap2 inhibited STAT5 signalling in HEK293T cells transfected with IL-7R$\alpha$ and the common gamma-chain ($\gamma$c) mediated by IL-7 at 10 and 100 pM (EC$_{50}$ = 41 pM) (Fig. 5h; Supplementary Fig. 7D,E).

To evaluate the potential of TSLP-traps to block TSLP-driven dendritic cell activation we quantified HLA-DR, CD40 and CD80 cell-surface expression levels and CCL17 chemokine production by primary human CD1c+ blood dendritic cells treated with TSLP alone[3,36] or in combination with antagonists (Fig. 6a,b). These experiments show that both TSLP-trap1 and TSLP-trap2 are able to significantly inhibit TSLP-driven DC activation, and that they are as potent in this regard as AMG-157.

**Structure and mechanism of TSLP antagonism by Tezepelumab.** We seized the opportunity to obtain structural and mechanistic insights into how Tezepelumab (AMG-157) might exert its antagonistic effects on TSLP and to characterize TSLP in a binary complex with a non-signalling binding partner, by pursuing the crystal structure of TSLP in complex with AMG-157$_{Fab}$. We were able to biochemically reconstitute and crystallize the TSLP$^{\Delta127–131}$: AMG-157$_{Fab}$ complex and to determine its crystal structure to 2.3 Å resolution (Fig. 7a; Table 1, Supplementary Fig. 8D). The structure reveals that the complementarity determining regions (CDRs) of the variable heavy chain domain (V$_H$) of AMG-157 target TSLP at the AB-loop region and C-terminal region of helix D, while the variable light chain fragment does not interact with TSLP at all (Fig. 7a). The TSLP:AMG-157 interface buries a total of 1,200 Å$^2$ of accessible surface area, with all three V$_H$-CDR loops contributing to a polar footprint (Fig. 7b; Supplementary Table 4). Most notably, Glu110 in the CDR-3 loop makes a bifurcated salt-bridge with TSLP-Arg150 and TSLP-Arg153, and Trp105 packs against TSLP-Cys75 in a surface pocket formed between TSLP $\alpha$D and the overhand AB-loop (Fig. 7b). Importantly, we now show that AMG-157 competes against a critical part of the TSLPR binding site on TSLP and remains completely clear of the IL-7R$\alpha$ binding site on the other side of the TSLP helical bundle (Fig. 7c). Furthermore, structural superposition of TSLP in its two complexes shows that the AB-loop and C-terminal tail extending from $\alpha$D adopt different conformations, with the rest of the TSLP main chain being very similar (r.m.s.d. of 0.56 Å for 89 aligned C$\alpha$ atoms). In fact the AB-loop and the C-terminal tail of $\alpha$D in the TSLP:AMG-157 complex are only partly resolved in the electron density maps, indicating that these regions are flexible in the absence of TSLPR consistent with our NMR and MD studies of TSLP (Figs 2f and 7d).

**Plasticity and functional role of the $\pi$-helix turn in TSLP.** The TSLP:AMG-157 complex provides a unique view of the IL-7R$\alpha$ binding site on TSLP in the absence of the shared receptor, thereby fuelling insights into the possible structural transitions associated with the cooperative recruitment of IL-7R$\alpha$ to the TSLP-mediated signalling complex. Perhaps the most intriguing feature of TSLP as bound to AMG-157 concerns an ordered water molecule that inserts into the $\pi$-helical turn of helix $\alpha$A in TSLP to provide a hydrogen-bonded bridge between the main-chain carbonyl and amide groups of Tyr43 and Lys49 (Fig. 7e). Such compensatory structural feature against the local disruption of the helix hydrogen bonding network has been linked to the stabilization of $\pi$-helical elements[46,55]. In accommodating the observed water molecule, the $\pi$-helical turn in $\alpha$A in the TSLP:AMG-157 complex is wider by about 1.5 Å than in the TSLP:TSLPR:IL-7R$\alpha$ complex (Fig. 7e). Thus, the $\pi$-helical turn in $\alpha$A is able to adopt at least two distinct conformational states. Given the localization of this part of $\alpha$A at the crossroads of the TSLP:TSLPR:IL-7R$\alpha$ complex (Fig. 2a) and its contribution to the IL-7R$\alpha$ binding epitope (Fig. 3e), we propose that the observed structural plasticity at the $\pi$-helical turn in $\alpha$A may be a key feature in the structural priming of the cytokine by TSLPR to enable high-affinity binding by IL-7R$\alpha$. In support of this notion and the functional role of receptor–receptor interactions in the ternary

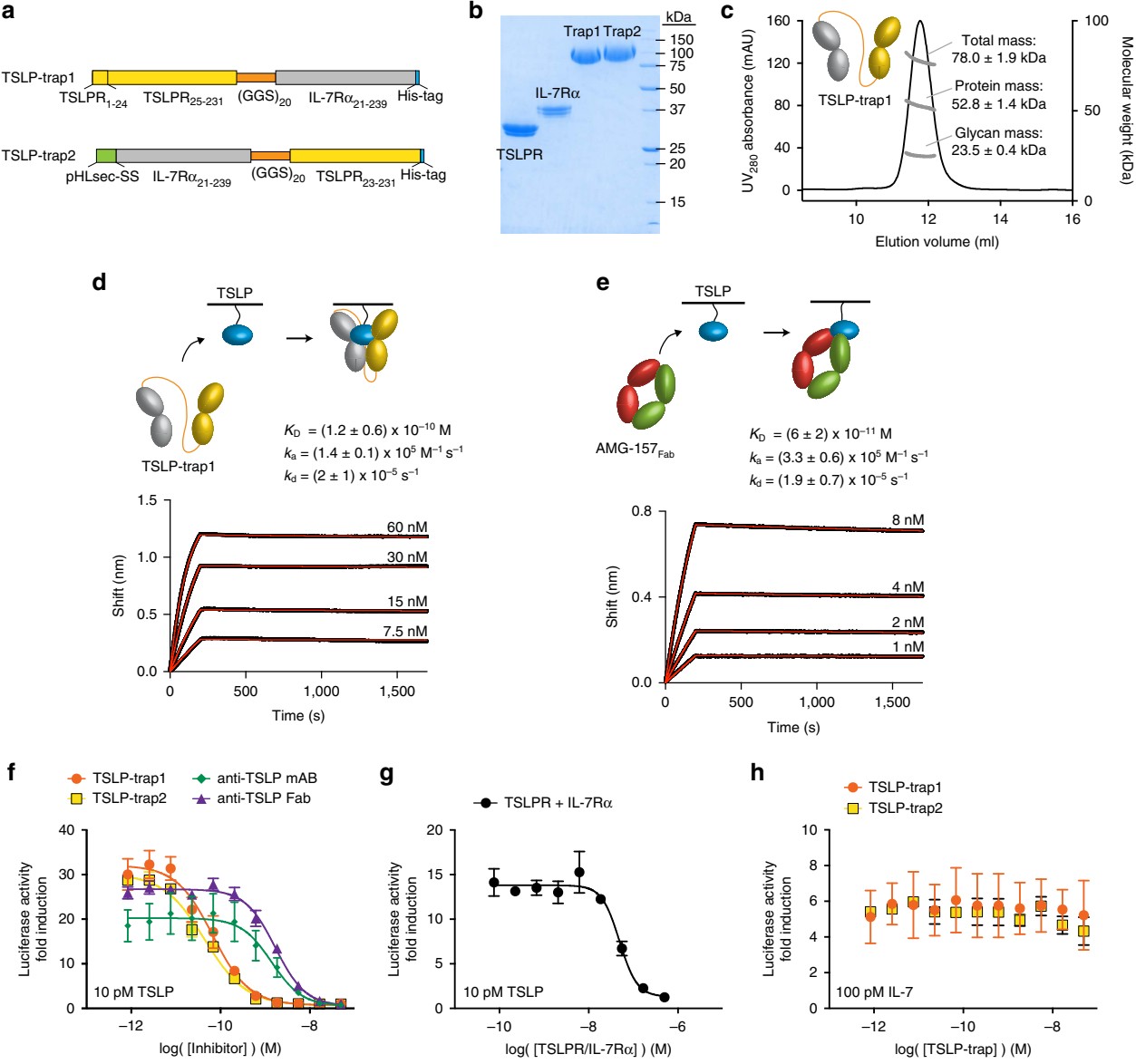

**Figure 5 | TSLP cytokine traps potently antagonize TSLP signalling *in vitro*.** (**a**) Design of TSLP-trap1 and TSLP-trap2 as fusion proteins between the IL-7Rα and TSLPR ectodomains interconnected with a (Gly-Gly-Ser)$_{20}$ linker. (**b**) Coomassie-stained reducing SDS–PAGE gel of purified TSLPR, IL-7Rα, TSLP-trap1 and TSLP-trap2. Molecular weights of protein standards are indicated. (**c**) SEC elution profile of TSLP-trap1 plotted as the ultraviolet absorbance at 280 nm (left *Y*-axis) in function of elution volume. The total, protein and glycan molecular weight (right *Y*-axis) as determined by MALLS are reported as the number average molecular mass and s.d. (**d,e**) BLI data traces (black) and the fitted 1:1 binding model (red) are plotted as the spectral nanometre shift in function of time for the interaction of TSLP-trap1 (**d**) and anti-TSLP AMG-157$_{Fab}$ fragment (**e**) with immobilized TSLP. The reported $K_D$-, $k_a$- and $k_d$-values represent average values and their s.d. from three technical replicate experiments. 4 nm of biotinylated TSLP was loaded for experiment (**d**) and 1 nm of TSLP was loaded for experiment (**e**). (**f**) STAT5 activity induced by 10 pM TSLP in function of increasing concentrations of inhibitors. The determined IC$_{50}$-values were 67 pM for TSLP-trap1; 44 pM for TSLP-trap2; 1.4 nM for AMG-157 and 1.7 nM for AMG-157$_{Fab}$. (**g**) STAT5 activity induced by 10 pM TSLP in the presence of increasing concentrations of an equimolar mixture of the soluble TSLPR and IL-7Rα ectodomains. The determined IC$_{50}$-value was 49 nM. (**h**) STAT5 activity induced by 100 pM IL-7 plotted in function of increasing concentrations of TSLP-traps. STAT5 activity in HEK293 cells is plotted as the luciferase activity fold induction. Each experiment was carried out in triplicate; data shown are averages, and error bars were calculated as s.e.m.

complex, IL-7Rα cannot be recruited to the TSLP:AMG-157 complex (Supplementary Fig. 8C).

Additional insights into the possible structural states of TSLP are provided by our NMR analysis. Close inspection of the $^1$H,$^{15}$N HSQC TSLP$^{Δ127-131}$ NMR spectrum at 900 MHz uncovered conformational heterogeneity on the second time-scale, which is much slower than can be sampled by MD-simulations. Specifically, we identified two populations for the Tyr43-Leu44-Ser45-Thr46 amide resonances located in the π-helical turn in αA of TSLP, as well as for the side-chain

resonance of Trp148, which stacks right above the π-helical turn of TSLP (Fig. 7f; Supplementary Fig. 9A–E). The minor conformations observed for Ser45 and Trp148 are populated to 20 ± 2% based on deconvoluted integrals of their respective signals, suggesting that they represent the same structural heterogeneity that connects the core of TSLP to αA. Although our NMR analysis does not provide structural details for the two TSLP conformations, together with the distinct conformational states of active versus antagonized TSLP, it provides independent support for the structural heterogeneity of TSLP.

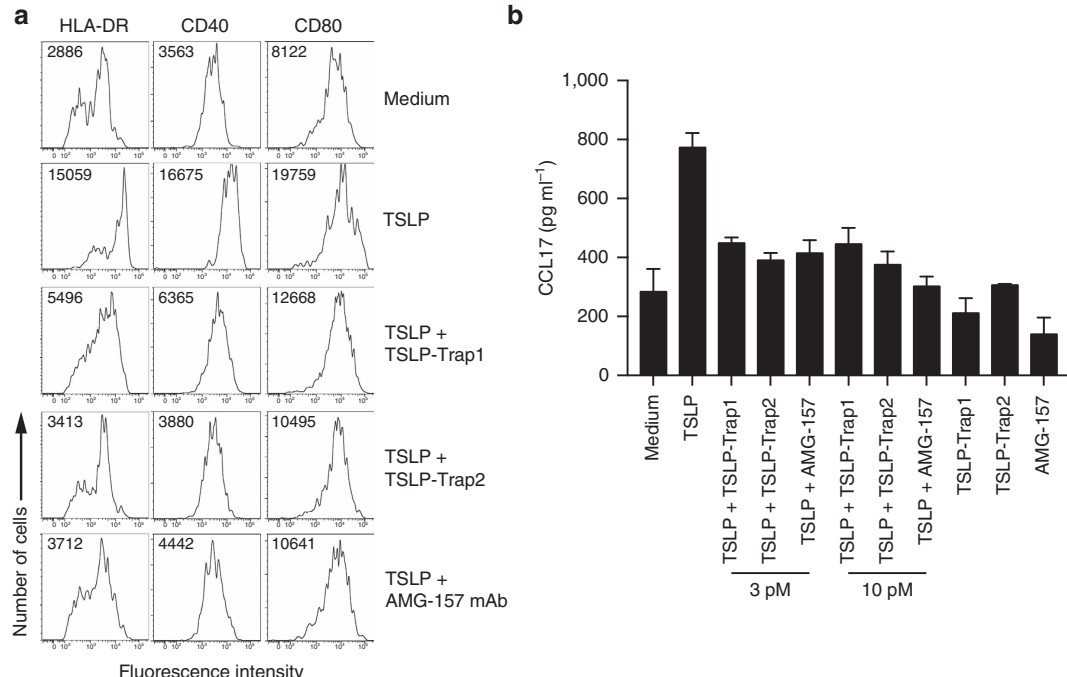

**Figure 6 | Effect of TSLP blockade on human blood dendritic cell maturation and chemokine production.** (**a**) Surface expression of different markers on CD1c$^+$ blood dendritic cells exposed to 10 pM TSLP in the presence or absence of 10 pM TSLP-Trap1, TSLP-Trap2 and AMG-157 mAb. (**b**) CCL17 production by CD1c$^+$ dendritic cells exposed to 10 pM TSLP in the presence or absence of different doses of TSLP-Trap1, TSLP-Trap2 and AMG-157 mAb. Medium and medium supplemented with TSLP antagonists (30 pM) were used as controls. Data are representative of two experiments. Bar graphs show the mean and s.e.m. error bars. Numbers indicate the mean fluorescence intensity.

## Discussion

The emergence of TSLP as a central orchestrator of Th2 responses that initiate allergy and inflammation has placed therapeutic targeting of TSLP-mediated signalling against major chronic diseases such as allergic asthma and atopic dermatitis at center stage. However, in order to maximally harness the therapeutic potential of TSLP-mediated signalling, it will be essential to dissect the structural and mechanistic basis for its bioactivity. Recent efforts that have leveraged mechanistic and structure-based considerations of cytokine-mediated receptor activation, have led to the development of engineered IL-2, IL-4 and IL-13 variants with drastically improved therapeutic efficacy and specificity, illustrating the power of consolidating such approaches in the development of tailor-made biologics[56–58]. In this study we have employed an integrative approach, including structural data at high-resolution, to propose a mechanistic blueprint for the activation and antagonism of the pro-inflammatory complex mediated by human TSLP.

The cornerstone of our mechanistic proposal is the high-affinity TSLP:TSLPR encounter complex driven by long-range electrostatic attraction of TSLP to its specific receptor at the cell surface, that primes two key concerted structural events: (i) allosteric activation of TSLP at site II by restructuring of its epicentre at the $\pi$-helical turn of $\alpha$A relayed by the structuring of the AB loop to enable recruitment of the shared receptor IL-7R$\alpha$ and (ii) positioning of the TSLPR membrane-proximal domain to facilitate interactions with the corresponding extracellular domain of IL-7R$\alpha$ (Fig. 8). The latter is potentially facilitated in part by the intrinsic, albeit low, affinity of the two receptors for each other, and partly by electrostatic attraction of IL-7R$\alpha$ to the TSLP:TSLPR binary complex, consistent with mechanistic implementation seen in diverse families of cytokine receptors[58–60].

The intrinsic cooperativity of the TSLP:TSLPR:IL-7R$\alpha$ complex is also the mechanistic pillar for the high *in vitro* potency of the

TSLP-traps we have developed by linking the TSLPR and IL-7R$\alpha$ ectodomains to create a single protein. Our TSLP-trap fusion proteins neutralize TSLP via a very drastic improvement in the $K_D$ compared to the unlinked counterparts by nearly three orders of magnitude ($K_D = 120$ pM) manifested primarily by very slow off-rate kinetics ($t_{1/2} \sim 10$ h). This strong increase in binding affinity is functionally reflected by the potent antagonistic activity and specificity of the TSLP-traps against TSLP signalling in our cellular inhibition assays with IC$_{50}$-values below 100 pM (Fig. 5). Such binding properties gain important biological context in light of the ability of both TSLP-trap1 and TSLP-trap2 to effectively antagonize TSLP-mediated molecular responses relevant for Th2 immunity in human primary cells (Fig. 6). Remarkably, the IC$_{50}$-values obtained for the TSLP-traps are 20–40-fold lower than those obtained for the AMG-157 mAb and its derived Fab fragment, which we propose is inextricably linked to the mechanistic modalities of the TSLP-mediated receptor complex (Fig. 8). Fusion proteins comprising receptor ectodomains and decoy receptors foster attractive binding properties to serve as effective therapeutics[61], as exemplified in the targeting of IL-1 (Rilonacept) or TNF$\alpha$ (Etanercept) for the treatment of CAPS-syndrome and rheumatoid arthritis, respectively[62]. Indeed, we are currently performing *in vivo* studies to assess the antagonistic potency of such fusion proteins. Furthermore, we anticipate that the current version of the TSLP-traps can be additionally improved in a number of ways, including optimizing linker length and introducing mutations to enhance the affinity and/or cross-linkage of the TSLPR and IL-7R$\alpha$ ectodomains to each other. Indeed, the therapeutic potential of targeting TSLP to treat allergic diseases mediated by Th2 responses is large, in particular in the context of combined approaches co-targeting the bioactivity of IL-25 and IL-33 (ref. 35).

Finally, our work on human TSLP provides opportunities to further investigate recent intriguing findings describing a second

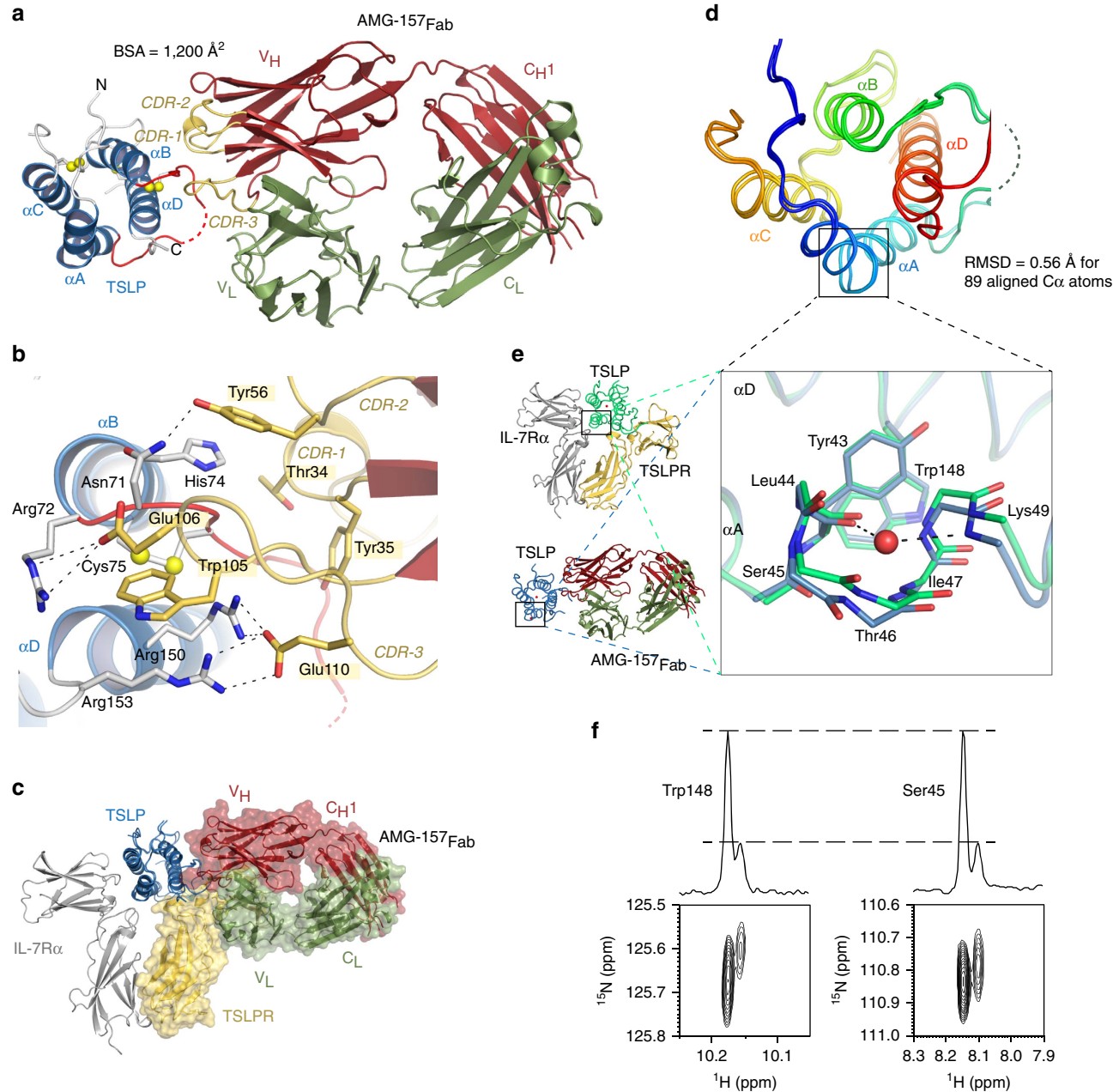

**Figure 7 | Structure of the TSLP:AMG-157_Fab complex and insights into TSLP antagonism and plasticity.** (**a**) Cartoon representation of the determined X-ray structure for AMG-157_Fab-fragment in complex with TSLP. TSLP helices (αA–αD) are coloured in blue and TSLP loop regions in white, except the *AB* loop (red). The red dashed line represents to two disordered residues in the TSLP *AB* loop. N- and C-termini are indicated. The V_H-C_H1 and V_L-C_L fragments of AMG-157_Fab are coloured in maroon and green, respectively. V_H CDR-loops are labelled and coloured yellow. BSA: buried surface area. (**b**) Detail of the TSLP:AMG-157_Fab interface. Selected interface residues are labelled and shown as sticks. Dashed lines represent salt bridges and hydrogen bonds. (**c**) Superposition of the TSLP:TSLPR:IL-7Rα and TSLP:AMG-157_Fab complexes based on the structural alignment of the two TSLP structures. TSLPR (yellow) and the light chain (green) and heavy chain (maroon) of the AMG-157_Fab fragment are displayed in surface mode. (**d**) Aligned human TSLP X-ray structures are shown as cartoons and are coloured according to a rainbow scheme with the N-terminus in blue and the C-terminus in red. (**e**) Structural comparison of the main chain conformation of the π-helical turn in TSLP helix A in both determined TSLP structures. The water molecule in the π-helical turn of helix αA of TSLP complex by AMG-157_Fab (blue) is shown as a red sphere. (**f**) Zoomed regions of the $^1$H,$^{15}$N HSQC-spectrum for TSLP$^{Δ127-131}$ at 900 MHz show the Nε–Hε Trp148 side chain (left) and Ser45 main chain amide (right) resonances. The projection of these regions on top of each panel shows that both minor forms adopt an identical population.

isoform of TSLP, termed short form TSLP (sfTSLP)[63–65]. Evidence for sfTSLP mainly pertains to the transcriptional levels of sfTSLP, and led to proposals that sfTSLP might be the constitutively expressed isoform of TSLP. sfTSLP is 63 residues in length and approximately covers the C-terminal half of human TSLP (residues 97–159). On our structure of human TSLP, sfTSLP would encompass αC, the long CD loop and αD (Supplementary Fig. 3A). It is presently unclear if sfTSLP can adopt any helical structure in the absence of αA and αB. However, a propensity to form amphipathic helices combined with a high isoelectric point (pI) of 11.1 would support its presumed function as antimicrobial peptide[63,66]. sfTSLP may also

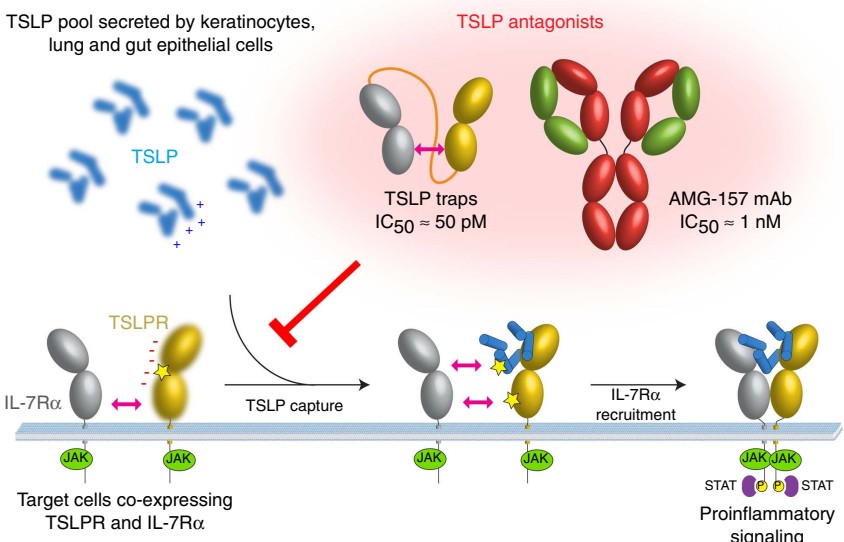

**Figure 8 | Mechanistic recapitulation for the assembly and antagonism of the pro-inflammatory TSLP mediated complex.** Following capture and rearrangement of TSLP by TSLPR at the cell surface, shared IL-7Rα is recruited to initiate intracellular pro-inflammatory JAK-STAT pathways. TSLP cytokine traps and anti-TSLP mAbs may represent effective strategies for therapeutic targeting of TSLP. Positive and negative signs represent the surface electrostatics of TSLP and TSLPR, respectively. Stars indicate site I, II and III. Left right arrows represent protein interactions.

exhibit anti-inflammatory properties[64,65] and may mediate immune tolerance in the gut[67].

We envisage that the structural and mechanistic framework and the molecular tools presented here will facilitate targeted interrogation of TSLP signalling *in vitro* and in animal models, and will guide therapeutic approaches that manipulate human TSLP-mediated signalling to treat allergic diseases.

## Methods

**Protein expression in mammalian cells and purification.** HEK293T (ATCC CRL-3216), HEK293S-TetR MGAT1$^{-/-}$ (ref. 41) and T-REx-293 (Thermo Fisher Scientific) cells were grown in high-glucose DMEM medium supplemented with 10% fetal calf serum, $10^6$ units per l penicillin G and $1\,g\,l^{-1}$ streptomycin in a 5% $CO_2$ atmosphere at 37 °C. The medium of T-REx-293 cells was supplemented with $5\,\mu g\,ml^{-1}$ blasticidin. Mammalian expression constructs for secreted proteins carrying a C-terminal hexahistidine-tag were generated in the pHLsec (ref. 68) and/or pcDNA4/TO vector (Thermo Fisher Scientific). For transient expression experiments 25 kDa branched PEI was used as transfection agent[68]. Before addition of the PEI-DNA transfection mix, the medium of confluently grown cells was changed to serum-free medium. Post transfection of 3–4 days, secretion of recombinant protein into the conditioned medium was confirmed by western blot analysis using an HRP-coupled antibody directed against the C-terminal His tag at 1:5,000 dilution ratio (Invitrogen, catalogue no. R931-25) and/or small-scale IMAC purifications in batch mode using 2 ml of conditioned medium.

Stable, tetracycline-inducible polyclonal cell lines for pcDNA4/TO expression constructs were generated in HEK293S-TetR MGAT1$^{-/-}$ cells or T-REx-293 cells by selection with $200\,\mu g\,ml^{-1}$ zeocin[42]. To induce expression the growth medium of confluently grown cells was replaced with serum-free medium supplemented with $2\,\mu g\,ml^{-1}$ tetracycline. For large-scale expression experiments, HEK293T and T-REx-293 cells were expanded to $850\,cm^2$ roller bottles and HEK293S-TetR MGAT1$^{-/-}$ cells to $175\,cm^2$ tissue-culture flasks or $145\,cm^2$ dishes. After 4–5 days following transient transfection or induction with tetracycline, the conditioned medium was harvested and clarified by centrifugation and filtration trough a $0.22\,\mu m$ bottle top filter. Recombinant proteins were captured from the clarified conditioned medium by IMAC purification using a cOmplete His-Tag purification column (Roche) and further purified by size-exclusion chromatography using preparative grade HiLoad 16/600 Superdex 75/200 columns (GE Healthcare) with HBS pH 7.4 as running buffer. Protein purity was evaluated on Coomassie-stained SDS–polyacrylamide gel electrophoresis (SDS–PAGE) gels (Figs 1f and 5b, Supplementary Figs 1A,8B and 10).

**Production of TSLP and IL-7 and soluble TSLPR and IL-7Rα.** Complementary DNA (cDNA) fragments (Genscript) encoding full-length human TSLP$^{R127A/R130S}$ (NP_149024.1; residue 1–159), human IL-7 (NP_000871.1; residue 1–177) and the extracellular fragments of human TSLPR (NP_071431.2; residue 1–221) and

human IL-7Rα (NP_002176.2; residue 1–239) were cloned into the pHLsec and pcDNA4/TO-expression vectors in frame with a C-terminal hexahistidine tag. The R127A and R130S mutations in TSLP remove a potential furin cleavage site[38]. For crystallization purposes, we also generated N-glycosylation mutants for TSLPR (TSLPR$^{N47Q}$, TSLPR$^{N47Q/N101Q}$ and TSLPR$^{N47Q/N169Q}$). Single-site N47Q, N101Q, N169Q mutants were ordered as synthetic genes from Genscript. The TSLPR$^{N47Q/N101Q}$ variant was created by overlap extension PCR using primers 1–4 (Supplementary Table 5). The TSLPR$^{N47Q/N169Q}$ was generated by restriction-based cloning. TSLPR-glycosylation variants were initially generated in the pHLsec-vector for transient expression and later subcloned in the pcDNA4/TO vector. Stable, tetracycline-inducible polyclonal cell lines for pcDNA4/TO expression constructs for the TSLP$^{R127A/R130S}$ and IL-7 cytokines, and the TSLPR (TSLPRWT, TSLPR$^{N47Q}$, TSLPR$^{N47Q/N101Q}$ and TSLPR$^{N47Q/N169Q}$) and IL-7Rα ectodomains were generated in HEK293S-TetR MGAT1$^{-/-}$ cells as described above.

**Production of biotinylated TSLP and soluble TSLPR and IL-7Rα.** To produce biotinylated versions of TSLP, TSLPR and IL-7Rα cDNA fragments for TSLP$^{R127A/R130S}$, and the TSLPR and IL-7Rα ectodomains were cloned between the EcoRI and KpnI sites of the pHL-AVITAG vector[68]. Before transfection in HEK293T cells, the culture medium was changed to serum-free DMEM medium supplemented with $100\,\mu M$ D-biotin. To allow specific C-terminal *in vivo* biotinylation, pHL-AVITAG constructs were co-transfected with the pDisplay-BirA-ER plasmid[69] in a 5:1 ratio. The conditioned medium was harvested five days after transfection and recombinant proteins were purified by IMAC and SEC.

**Production of recombinant TSLP and IL-7Rα in *E. coli*.** cDNA fragments (Genscript) encoding human TSLP$^{\Delta127-131}$ (NP_149024.1; residue 29–159) and the extracellular fragment of human IL-7Rα (NP_002176.2; residue 21–239) were cloned in the pET15b expression vector in frame with a cleavable N-terminal hexahistidine tag. TSLP$^{\Delta127-131}$ and IL-7Rα were expressed in the *E. coli* BL21(DE3) strain as inclusion bodies and refolded *in vitro*[40] using 6 M Guanidine-HCl as denaturing agent. Following refolding, the N-terminal His-tag was removed using biotinylated thrombin (Novagen). Biotinylated thrombin was removed by incubation with streptavidin agarose beads. Refolded proteins were further purified by size-exclusion chromatography using a Superdex 75 column with HBS pH 7.4 as running buffer.

**Preparation of TSLP:TSLPR:IL-7Rα complex for crystallization.** Following purification of TSLPR$^{N47Q}$ from stable HEK293S-TetR MGAT1$^{-/-}$ cells, N-linked glycosylation was trimmed by overnight incubation at room temperature with EndoH (New England Biolabs) using 7.5 kU of EndoH per mg of complex. The binary TSLP:TSLPR$^{N47Q}$ complex was formed by adding a molar excess of refolded TSLP$^{\Delta127-131}$ to EndoH-treated TSLPR$^{N47Q}$. The binary complex TSLP$^{\Delta127-131}$:TSLPR$^{N47Q}$ was isolated and separated from the excess of TSLP$^{\Delta127-131}$ by SEC using a Superdex 75 column with HBS pH 7.4 as running buffer. The ternary complex was then formed by adding a molar excess of the refolded IL-7Rα ectodomain

to the binary TSLP$^{\Delta127-131}$:TSLPR$^{N47Q}$ complex. The ternary complex was isolated and separated from the excess of IL-7Rα by SEC using a Superdex 200 column with HBS pH 7.4 as running buffer. Fractions corresponding to the ternary TSLP$^{\Delta127-131}$:TSLPR$^{N47Q}$:IL-7Rα complex were pooled and concentrated by centrifugal ultra-filtration to a concentration of 6 mg ml$^{-1}$. The protein sample was then aliquoted and flash frozen in liquid nitrogen.

**Production of anti-TSLP mAb and Fab fragment and Fab:TSLP.** cDNA fragments (Gen9) encoding the Tezepelumab (AMG-157) lambda light chain, the IgG2 heavy chain[54], and the PCR-derived V$_H$-C$_H$1 heavy chain fragment (primers 5 and 6) (Supplementary Table 5) were cloned between the AgeI and KpnI sites of the pHLsec vector, in frame with the vector's signal peptide. At the C-terminus, the heavy chain and the V$_H$-C$_H$1 fragment carried a hexahistidine tag. The mAb and its Fab-fragment were produced by co-transfecting HEK293T cells with expression plasmids for the light chain and, heavy chain or V$_H$-C$_H$1-fragment in a 1:1 ratio. The mAb or Fab-fragment were purified from the conditioned medium by IMAC (Roche cOmplete column) and SEC (Superdex 200) using HBS pH 7.4 as running buffer. The Fab:TSLP complex was formed by incubating the Fab-fragment with a molar excess of refolded TSLP$^{\Delta127-131}$ produced in *E. coli* as described above. The complex was then isolated from the molar excess of TSLP$^{\Delta127-131}$ by SEC and concentrated to 10 mg ml$^{-1}$. The protein sample was then aliquoted and flash frozen in liquid nitrogen.

**Protein crystallization.** Nanoliter-scale vapour diffusion crystallization experiments were set up at 293 K using a Mosquito crystallization robot (TTP Labtech) and commercially available sparse-matrix screens (Molecular Dimensions, Hampton research). The TSLP$^{\Delta127-131}$:TSLPR$^{N47Q}$:IL-7Rα ternary complex crystallized in condition H5 of the PEG/Ion HT screen (0.02 M Citric acid, 0.08 M BIS-TRIS propane, pH 8.8, 16% w/v Polyethylene glycol 3,350). Following gradient optimization, crystals were cryoprotected by a 1 min soak into mother liquor supplemented with 20% ethylene glycol. The TSLP:AMG-157$_{Fab}$ complex crystallized in condition B8 of CrystalScreen HT (0.2 M ammonium sulfate, 0.1 M sodium acetate pH 4.6 and 25% w/v polyethylene glycol 4,000) and crystals were cryo-protected with mother liquor supplemented with 20% v/v polyethylene 400. Crystals were cryo-cooled by direct plunging into liquid nitrogen.

**Crystallographic structure determination.** X-ray diffraction measurements were conducted at the Proxima2A beam line (synchrotron SOLEIL, Gif-sur-Yvette, France). All data were integrated and scaled using the XDS suite[70]. The structures for the TSLP$^{\Delta127-131}$:TSLPR$^{N47Q}$:IL-7Rα complex and the TSLP$^{\Delta127-131}$:AMG-157$_{Fab}$ complex were determined by maximum-likelihood molecular replacement (MR) as implemented in the program suite PHASER (ref. 71). Human TSLP and TSLPR search models were derived from an X-ray structure for human TSLP:TSLPR in complex with mouse IL-7Rα (PDB entry 5J12, to be published) that was also solved by MR using search models derived from the mouse TSLP:TSLPR:IL-7Rα complex (PDB entry 4NN5)[37]. The search model for human IL-7Rα was obtained from the human IL-7:IL-7Rα complex (PDB entry 3DI2)[51]. As a search model for the AMG-157$_{Fab}$ light chain, chain B of PDB entry 4HK0 was used, and for the AMG-157 heavy chain fragment, chain A of PDB entry 4HIE was used. Both crystal forms contained one complex in their asymmetric unit. Model (re)building was performed in COOT (ref. 72) and individual coordinate and ADP refinement (combined with TLS parameterization for the ternary TSLP:TSLPR:IL-7Rα complex) was performed in PHENIX (ref. 73) and autoBuster[74]. Model and map validation tools in COOT, the PHENIX suite and the PDB_REDO server[75] were used throughout the work flow to guide improvement and validate the quality of crystallographic models.

**Disulfide determination in TSLPR.** Following digestion of iodoacetamide-treated TSLPR ectodomain with trypsin (sequencing grade modified trypsin, Promega, V511), the peptide mixture was spotted onto an Opti-TOF 384 Well MALDI Plate (ABsciex, Framingham, MA 01,701, USA, PN 10,16,629), in a 1:1 ratio mix with MALDI matrix α-cyano-4-hydroxy cinnamic acid (Sigma, 4,76,870), prepared in a concentration of 5 mg ml$^{-1}$ in 0.1% trifluoroacetic acid, 10 mM ammonium citrate, 50% Acetonitrile. Sample MS and fragmentation spectra were acquired on a 4,800 Proteomics Analyzer, a MALDI-TOF-TOF instrument (ABsciex, Framingham, MA 01,701, USA), using the delayed extraction and reflector technologies in the positive ion mode. Default settings and factory acquisition methods were used. The instrument was calibrated with Glu-Fibrinopeptide standards (Applied Biosystems 4,700 Proteomics Analyser Mass Standards kit, ABSciex, 43,33,604).

**Design and production of TSLP-traps.** The extracellular domains of human TSLPR (NP_071431.2; residues 1–231) and human IL-7Rα (NP_002176.2; residues 1–239) were amplified by PCR using primer pairs 7–8 (TSLPR) and 9–10 (IL-7Rα), respectively (Supplementary Table 5). The PCR fragments were cloned into the EcoRI/XbaI opened pEF6-myc/HisA expression vector in frame with a C-terminal myc/hexahistidine tag, resulting in pEF-ShTSLPR and pEF-ShIL7Rα. Human TSLP-trap1 (pEF-hTSLPtrap1) was generated by PCR amplification of a (GGS)$_{20}$-linker fragment from a plasmid template (primer pair 11–12) and the extracellular

part of the human IL-7Rα (residues 21–239) with the C-terminal myc/His tag from pEF-ShIL7Rα with primer pair 13–14 (Supplementary Table 5). Both fragments were ligated in frame by a 3-point ligation into the XbaI/PmeI opened pEF-ShTSLPR vector. In the resulting fusion construct the human IL-7Rα extracellular domain with C-terminal myc/His tag is connected by the (GGS)$_{20}$-linker fragment to the C-terminus of the human soluble TSLPR. Human TSLP-trap2 (pEF-hTSLPtrap2) was generated likewise by ligating the same linker fragment together with that encoding the extracellular domain of human TSLPR (residues 23–231 and C-terminal myc/His tag, PCR amplified from pEF-ShTSLPR with primer pair 14–15) into the XbaI/PmeI opened pEF-ShIL7Rα vector (Supplementary Table 5). Finally, the natural signal sequence of hIL7Rα was replaced by subcloning the open reading frame of hTSLPtrap2 (starting with residue E21 of the mature hIL7Rα sequence) by PCR-cloning (primer pair 16–17) (Supplementary Table 5) into EcoRI/PmeI of pEF6-ssFlag which contains the signal sequence of the murine IL-33Rα followed by a Flag-tag. The final expression vectors were generated by cloning the cDNA-fragments encoding TSLP-trap1 and TSLP-trap2 into the multicloning site of the the pcDNA4/TO-expression vector (Thermofisher) in frame with a C-terminal hexahistidine tag using primer pairs 18–19 and 20–21 (Supplementary Table 5). For TSLP-trap1, the native secretion signal of human TSLPR was used, while for TSLP-trap2 the signal peptide from the pHLsec-vector was used[68]. Stable, inducible cell lines for TSLP-trap1 and TSLP-trap2 were generated in T-REx-293 cells as described above.

**SEC-MALLS.** Protein samples (100 μl) were injected onto a Superdex 200 Increase 10/300 GL column (GE Healthcare), with HBS pH 7.4 as running buffer at 0.5 ml min$^{-1}$, coupled to an online ultraviolet-detector (Shimadzu), a multi-angle light scattering miniDAWN TREOS instrument (Wyatt) and a Optilab T-rEX refractometer (Wyatt) at 25 °C. A refractive index increment (dn/dc) value of 0.185 ml g$^{-1}$ was used for protein concentration and molecular mass determination. Data were analysed using the ASTRA6 software (Wyatt). Correction for band broadening was applied using parameters derived from BSA injected under identical running conditions. For the analysis of TSLP-traps, conjugate analysis was performed using theoretical protein extinction coefficients and a dn/dc-value of 0.160 ml g$^{-1}$ for the glycan modifier.

**Biolayer interferometry.** BLI experiments were performed in PBS-buffer supplemented with 0.01% (w/v) BSA and 0.002% (v/v) Tween 20, with an Octet RED96 instrument (FortéBio), operating at 25 °C. Streptavidin-coated biosensors were functionalized with biotinylated TSLP$^{R127A/R130S}$, TSLPR or IL-7Rα and quenched with a 10 μg ml$^{-1}$ biotin solution and then dipped into solutions containing different analyte concentrations. IL-7, TSLPR and IL-7Rα ectodomains produced from stable transfected HEK293S-TetR MGAT$^{-/-}$ cells were used as analyte. To verify that no non-specific binding was present during the interaction assay, non-functionalized biosensors were used as a control. To measure the interaction between IL-7Rα and the TSLP:TSLPR complex, TSLP-loaded sensor tips were incubated with 320 nM of TSLPR which was also included in the assay buffer and all IL-7Rα samples. The sensor traces from zero concentration samples were subtracted from the raw data traces before data analysis. To correct for bulk effects during the measurements for the interaction between IL-7Rα and TSLPR a column of non-functionalized sensors was used to enable double reference subtraction. All data were fitted with the FortéBio Data Analysis 9.0.0.4 software using a 1:1 ligand model.

**Small-angle X-ray scattering data collection and analysis.** SAXS data were measured on the SWING beam line at the SOLEIL Synchrotron (Gif-sur-Yvette, France). Around 50 μl of glycan-minimized ternary TSLP$^{\Delta127-131}$:TSLPR$^{N47Q}$:IL-7Rα complex (6 mg ml$^{-1}$), as prepared for crystallographic studies, was injected onto an Agilent 4.6 × 300 mm Bio SEC-3 column with 300 Å pore size and HBS pH 7.4 as running buffer at a flow speed of 0.2 ml min$^{-1}$ at 288 K. X-ray scattering data were collected in continuous flow mode with 1 s exposure time per frame and an acquisition rate of 1 frame every 2 s. Data were recorded within a momentum transfer range of 0.01 Å$^{-1}$ < q < 0.6 Å$^{-1}$, with $q = 4\pi\sin\theta/\lambda$. Raw data were radially averaged and buffer subtracted using Foxtrot v3.2.7 (developed at Synchrotron SOLEIL and provided by Xenocs (Sassenage, France)). The quality of the data was analysed with Foxtrot by checking the stability of the radius of gyration over the length of the elution peak and by scaling all curves to the most intense scattering profile. The final scattering curve was obtained by averaging the unscaled, buffer-subtracted scattering profiles from frames 119–128, which correspond to the top of the elution peak. Structural parameters were determined with the ATSAS suite[76].

**Constructs for cellular activity assays and binding studies.** The pMET7-TSLP$^{R127A/R130S}$-SEAP-Flag allows the expression of a human TSLP-secreted alkaline phosphatase fusion protein (TSLP-SEAP). pMET7-TSLP$^{R127A/R130S}$-SEAP-Flag was created by ligating a codon optimized cDNA fragment for TSLP$^{R127A/R130S}$ with a C-terminal GGSGGS linker into the EcoRI/BglII opened pMET7-CRH2-SEAP-Flag vector using primers 22 and 23 (Supplementary Table 5). pMET7-Flag-TSLPR and pMET7-HA-IL-7Rα allow the expression of full-length FLAG-tagged human TSLPR and HA-tagged human IL-7Rα. pMET7-Flag-TSLPR was created by ligating

a codon optimized hTSLPR cDNA-fragment into the ClaI/XbaI opened pMet7-flag-mTSLP vector[37]. pMET7-HA-IL-7Rα was created by ligating a codon optimized hIL-7Rα cDNA fragment into the BspEI/XbaI opened pMet7-HA-mouseIL7Rα vector[37]. Site-directed mutations in these vectors were introduced via the Quickchange protocol (Stratagene). Site-directed mutations of pHL-hTSLP$^{R127A/R130S}$ were first introduced in the pUC57-hTSLP vector via the Quickchange protocol, followed by ligation of the EcoRI/KpnI mutant hTSLP DNA fragment into the EcoRI/KpnI opened pHL-hTSLP$^{R127A/R130S}$ vector[37]. All primers used for site-directed mutagenesis of human TSLP, TSLPR and IL-7Rα are provided in Supplementary Table 6.

**Competitive TSLP-SEAP cell binding assay.** HEK293T cells were transfected with pMET7-TSLP$^{R127A/R130S}$-SEAP using linear PEI (Polysciences). The culture medium was replaced with Optimem medium (Life technologies) one day after transfection, and the medium containing secreted TSLP-SEAP fusion protein was harvested three days after transfection. For expression of human TSLPR, HEK293T cells were transfected with 0.875 ng pMet7 vector and 0.125 ng pMet7-FLAG-TSLPR per well in 6-well plates using linear PEI (Polysciences). Two days after transfection, the cells were detached with 5 mM EDTA in phosphate buffered saline (Life Technologies) and were washed in FACS buffer (1% fetal bovine serum, 0.5 mM EDTA in phosphate buffered saline). Subsequently, 130,000 cells were incubated for 2 h at 6 °C with TSLP-SEAP containing conditioned medium (diluted 15-fold) and different concentrations of unlabelled wild type or mutant TSLP in FACS buffer. The concentration of wild type and mutant TSLP was determined by ELISA (Human TSLP Duoset ELISA, R&D Systems). The cells were washed three times with FACS buffer, and were used to quantify the amount of bound alkaline phosphatase activity using the PhosphaLight kit (Tropix) in an Envision chemi-luminescence counter (Perkin-Elmer). The data were plotted and fitted to a log inhibitor versus response curve as implemented in Graphpad Prism.

**TSLP induced STAT5 reporter activation.** For comparing wild type and mutant TSLP, HEK293T cells were co-transfected with 15 ng pMET7-Flag-TSLPR, 15 ng pMET7-HA-IL-7Rα, 900 ng empty pMET7 vector and 100 ng pGL3-β-casein-luci reporter plasmid per well of a 6-well plate. When comparing wild type and mutant receptors, HEK293T cells were co-transfected with 150 ng pMET7-Flag-hTSLPR, 150 ng pMET7-HA-IL-7Rα, 600 ng empty pMET7 vector and 100 ng pGL3-β-casein-luci reporter plasmid per well of a 6-well plate. The pGL3-β-casein-luci luciferase reporter carries a set of five repeated STAT5-responsive motifs of the β-casein promoter. One day after transfection, the cells were detached with cell dissociation buffer (Life Technologies), and resuspended in DMEM + 10% fetal bovine serum. Following counting, 50% of the cells were seeded in a new six-well plate for FACS analysis, and 2% of the cells were seeded per well in 96 well plates and stimulated with increasing concentrations of hTSLP. The luciferase activity was determined on day two after transfection using an Envision chemiluminescence counter. The fold induction of luciferase activity was calculated by the ratio of the luminescence signal (cps) from cells stimulated with hTSLP to the signal from the unstimulated cells. The data were plotted and fitted to a log agonist versus response curve in Graphpad Prism.

The expression of FLAG-tagged hTSLPR at the cell surface was determined using a mouse monoclonal anti-FLAG M2 antibody (Sigma) and Alexfluor488 labelled goat anti-mouse antibody (Molecular Probes) on a FACSCalibur (BD Biosciences). HA-tagged hIL-7Rα expression was determined using a FITC-labelled mouse monoclonal anti-HA antibody (Sigma). A gate was set that distinguishes between cells with low (background) fluorescence and increased fluorescence. Only 'gated' cells with increased fluorescence were used to calculate receptor expression levels. Relative receptor expression was determined as number of gated cells multiplied by the mean fluorescence of the gated cells.

**IL-7 induced STAT5 reporter activation.** HEK293T cells were co-transfected with 1,000 ng pREX-IRES-CD4-gamma common, 2 ng pMET7-HA-IL-7Rα, 200 ng pMX-IRES-GFP-hJak3, 133 ng empty pMET7 vector and 100 ng pGL3-β-casein-luci reporter plasmid per well of a 6-well plate. One day after transfection, cells were detached and seeded in 96 well plate as described above and incubated overnight with human IL-7. Luciferase activity was measured one day later as described above. The pREX-IRES-CD4-gamma common and pMX-IRES-GFP-hJak3 vectors[77] were kindly provided by Dr S.N. Constantinescu (Ludwig Institute for Cancer Research, Belgium).

**Inhibition in the TSLP induced STAT5 reporter assay.** To study the effect of different inhibitors (TSLP-traps, receptor ectodomains, anti-TSLP AMG-157 mAb or derived Fab fragment) on TSLP induced STAT5 reporter assays, HEK293T cells were seeded and transfected as described above. The day after transfection, the cells were detached with cell dissociation buffer (Life Technologies), and resuspended in DMEM + 10% fetal bovine serum. Approximately 3% of the cells were seeded in 50 μl medium per well in 96 well plates. In a separate plate, TSLP produced from HEK293S-TetR MGAT$^{-/-}$ cells was incubated in medium with increasing concentrations of the inhibitors for 30 min at room temperature. After this pre-incubation, 50 μl of this TSLP-inhibitor mix was added to the seeded cells. The reported concentrations for inhibitor and TSLP are their final concentrations in

this 100 μl volume. Cells were incubated overnight with this mixture and STAT5 reporter luciferase activity was measured 24 h after the start of the stimulation. Fold induction of luciferase activity was calculated by dividing the luminescence signal (counts per second) of the TSLP stimulated cells by the luminescence signal of the unstimulated cells. The data were fitted to a log inhibitor versus response curve in GraphPad Prism.

**Inhibition of dendritic cell activation by TSLP antagonists.** CD1c$^+$ dendritic cells (DCs) were purified from adult blood buffy coats (Red Cross Flanders, Belgium). Peripheral blood mononuclear cells (PBMC) were separated by Ficoll centrifugation. Cells were then depleted from CD19$^+$ B cells using magnetic beads (Miltenyi Biotec). The negative fraction was then enriched for CD1c$^+$ dendritic cells by labelling them with anti-CD1c biotinylated antibodies (1:15 dilution), followed by anti-biotin microbeads (Myltenyi Biotec). CD1c$^+$ DCs were cultured immediately after purification in RPMI containing 10% fetal calf serum (FCS), and penicillin-streptomycin. Cells were seeded at $0.5 \times 10^6$ per ml in flat-bottomed 96-well plates in the presence of E. coli-derived human TSLP at 10 pM, or TSLP-trap1, TSLP-trap2 or AMG-157 mAb at 3 and 10 pM. As controls, medium or medium supplemented with TSLP-trap1, or TSLP-trap2, or AMG-157 mAb at 30 pM were included. After 24 h in culture, DCs were collected and stained with anti-human CD40, CD80 and HLA-DR (all from BD Biosciences). Cells were analysed with a LSRII Fortessa flow cytometer (BD Biosciences). Dead cells were excluded based on DAPI positivity. Results were analysed with FlowJo software. DC culture supernatants were collected at 24 h, and analysed for the presence of CCL17 using a specific ELISA (R&D Systems). Endotoxins were removed from recombinant proteins with ε-poly-L-lysine spin columns (Pierce). Resulting endotoxin-levels were determined with an Endosafe-PTS system (Charles River) as lower than 5–8 EU mg$^{-1}$ recombinant protein.

**Nuclear magnetic resonance.** Isotopically labelled $^{15}$N-TSLP$^{Δ127-131}$ and $^{13}$C/$^{15}$N-TSLP$^{Δ127-131}$ were produced in E. coli BL21(DE3) cells transformed with the pET15b-TSLP$^{Δ127-131}$ expression construct (see above). Cells were grown in minimal medium at 37 °C supplemented with a $1 \times$ MEM vitamin solution (Sigma Aldrich, M6895), 1 g l$^{-1}$ $^{15}$NH$_4$Cl (Sigma Aldrich, 2,99,251), 3.5 g l$^{-1}$ U-$^{13}$C6-glucose (EURISO-TOP, CLM-1396) and induced with 1 mM IPTG. Isotopically labelled TSLP$^{Δ127-131}$ was refolded from inclusion bodies and its N-terminal His-tag was removed as described above. Protein samples for NMR measurements at concentrations of 582 μM (8.5 mg ml$^{-1}$) for $^{15}$N-TSLP$^{Δ127-131}$ and 628 μM (9.2 mg ml$^{-1}$) for $^{13}$C/$^{15}$N-TSLP$^{Δ127-131}$ were prepared in 20 mM NaH$_2$PO$_4$ pH 6.8, 50 mM NaCl, 2.5 mM EDTA and $1 \times$ cOmplete Protease Inhibitor cocktail. NMR spectra were recorded on 600 MHz Bruker and 900 MHz US$^2$ Bruker NMR spectrometer instruments at the CNRS Structural and Functional Glycobiology Unit (Parc Scientifique de la Haute Borne, Lille, France) and assignment of the triple resonance NMR spectra of TSLP was obtained by the product-plane approach[78]. Assignment of the tryptophan side chains was based on the NOE contact between the tryptophan amide resonance previously assigned by triple resonance spectroscopy and the Hδ side chain proton of tryptophan.

**Molecular dynamics simulations.** *TSLP conformational plasticity.* The conformational plasticity of TSLP and TSLP bound to TSLPR was investigated by molecular dynamics simulations. As X-ray structures for TSLP do not provide any density in the loop region spanning from residues 115–132, fifty models were generated for this region using Modeller 9.14 (ref. 79). To account for the structural heterogeneity of this loop, five diverse loop models were selected for molecular dynamics simulations. The apo-TSLP structures and TSLP:TSLPR complex structures were prepared separately. Five 250 ns molecular dynamics simulations, each with a different TSLP starting model, were performed for TSLP and TSLP:TSLPR (10 runs in total). All simulations were performed using Gromacs 5.1.1 (ref. 80) with the Amber99SB-ILDN force field and TIP3P explicit solvent. The crystal structure was placed in a rhombic dodecahedron extending 1.2 nm beyond the diameter of the system. An integration time step of 2 fs and the Verlet scheme were used for all simulations. Van der Waals and short-range Coulomb forces were truncated to 10 Å. Long-range Coulomb forces were treated with the particle mesh Ewald method and bonds involving hydrogen atoms were constrained. During equilibration, protein heavy atoms were harmonically restrained with a force constant of 1,000 kJ mol$^{-1}$ nm$^{-1}$. The crystal structure was relaxed using a steepest descent algorithm until the maximum force exerted on any atom was lower than 1,000 kJ mol$^{-1}$. A 300 ps NVT equilibration was then performed, starting at 30 K and increasing the temperature to 300 K over the course of 200 ps. Temperature control was achieved through two Bussi–Parinello thermostats coupled to protein and non-protein groups, each with a coupling time of 0.1 ps. Following NVT equilibration the system was coupled to a Berendsen barostat with a reference pressure of 1 bar and a coupling time of 0.5 ps for 500 ps of NPT equilibration. The 250 ns long production runs were performed using two Nose–Hoover thermostats with coupling times of 1 ps and reference temperatures of 300 K. Pressure control was achieved through a Parinello-Rahman barostat with a reference pressure of 1 bar and a coupling time of 2 ps. Snapshots were saved every 100 ps. Root-mean-square fluctuations around the average structure were calculated for the final 100 ns of simulation time.

*Water-stripped TSLP.* An interesting feature of the TSLP crystal structure is the presence of a buried water molecule in the core. During the MD simulations described above, the buried water molecule remained bound in the protein core. To assess its structural role further, molecular dynamics simulations were performed in which the central water molecule was deleted from the starting model. Three 250 ns all-atom molecular dynamics simulations were completed. Spontaneous rehydration of this cavity through a channel located between the B and C helices by bulk water molecules was observed within 15–125 ns in each of three independent simulations.

*Inserted water molecule at the π-helical turn of TSLP helix A.* A stabilizing water molecule can be observed in the A helix of the TSLP:AMG-157$_{Fab}$ crystal structure. We sought to investigate if water molecules were present at a similar position in our simulations by identifying water molecules for which the distance between the water oxygen, and the Tyr43 carbonyl oxygen and the Lys49 amide nitrogen was equal to or less than 3.5 Å. Such water molecules were identified in 19% of frames over the entire TSLP simulations.

**Data availability.** Protein Data Bank: Coordinates and structure factors for the crystal structure of the TSLP$^{\Delta127-131}$:TSLPR$^{N47Q}$:IL-7Rα complex and TSLP$^{\Delta127-131}$:AMG-157$_{Fab}$ complex have been deposited with accession codes 5J11 and 5J13, respectively. Other PDB codes used in this study: 5J12, 4NN5, 3DI2, 4HIE, 4HK0 and 3UP1.

Small Angle Scattering Biological Data Bank: SAXS data and coordinates of the best TSLP$^{\Delta127-131}$:TSLPR$^{N47Q}$:IL-7Rα model generated by the AllosMod-FoXS server have been deposited with accession code SASDB99.

Protein sequences used in this study: Thymic stromal lymphopoietin (TSLP): NCBI NP_149024.1; Thymic stromal lymphopoietin receptor (TSLPR): NCBI NP_071431.2 and Uniprot ID Q9HC73; Interleukin-7 (IL-7): NCBI NP_000871.1; Interleukin-7 receptor α (IL-7Rα): NCBI NP_002176.2 and Uniprot ID P16871; Secreted alkaline phosphatase (SEAP): Uniprot P05187; Bovine Serum Albumine (BSA): Uniprot ID P02769; Bifunctional ligase/repressor BirA (BirA) UniprotID: L3K9G4. All other data are available from the corresponding author on reasonable request.

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

## Acknowledgements

We thank the staff of Proxima 2A and SWING beam lines at SOLEIL synchrotron for excellent technical support; K.V. and K.P. are postdoctoral fellows of Research Foundation Flanders, Belgium (FWO). D.V.R. is a PhD fellow of the Research Foundation Flanders, Belgium (FWO). This work was supported by grants from the FWO (no G0C2214N to S.N.S. and F.P.), the Hercules Foundation (no AUGE-11–029 to S.N.S.), the Belspo 'Interuniversity Attraction Poles' (no P7/32 to R.B.), the Ghent University 'Concerted Research Actions' (no BOF13-GOA-005 to R.B.), the Ghent University 'Group-ID Multidisciplinary research partnership' (to R.B.). The 900 MHz NMR facility was supported by the CNRS (TGIR RMN THC, FR-3,050, France), University of Lille 1, the European community (EDRF), and the Région Nord-Pas de Calais (France). Computational resources and services for MD calculations were provided by the VSC (Flemish Supercomputer Center), funded by the FWO and the Flemish Government—department EWI. We thank Koen Verschueren for carefully reading the manuscript.

## Author contributions

K.V. and A.D. cloned expression constructs, performed protein expression and purification and crystallization experiments. K.V. performed MALLS and SAXS experiments. K.V. and S.N.S. determined crystallographic structures and carried out structural analyses. J.L. and G.L. performed NMR studies. K.V. performed BLI experiments with contributions from KP. F.P. designed and performed cellular competition and activity assays with contributions from J.T. F.P. and A.D. carried out site-directed mutagenesis. H.B. and R.B. designed and initiated production of the TSLP traps. I.V. performed mass spectrometry. D.V.R. carried out MD simulations with contributions from H.D.W.. H.H. and B.N.L. performed cellular assays in primary human cells. K.V. and S.N.S. wrote the manuscript with contributions from all authors. S.N.S. conceived and supervised the project.

## Additional information

**Competing interests:** K.V., F.P., H.B., R.B. and S.N.S. have filed a patent application with the European Patent Office (EP 1,61,63,883.8) pertaining to the development of the TSLP-traps. The remaining authors declare no competing financial interests.

