## [Peer review file · Nature Communications]

Reviewers' comments:

Reviewer #1 (Remarks to the Author):

this paper reports the crystal structure of the human TSLP complex, as well as accompanying protein engineering studies to antagonize TSLP involving both antibody and mechanism based approaches. The work is important for establishing a sound foundation for TSLP based drug therapy and should be of interest to readers of Nature Communication. All of the work is technically sound and I have no reservations in recommending acceptance.

Reviewer #2 (Remarks to the Author):

S. Savvides and co-workers present a comprehensive description of a ternary complex involved in the pathophysiology of allergic diseases including asthma and atopic dermatitis. Thus, the results may guide future therapeutic approaches to treat such widespread allergic diseases.

From this structure they determine important residues at the interfaces of the subunits and strengthen these observations with a large number of mutational studies. Furthermore, they shed light on the antagonistic mechanism of a therapeutic antibody which is already in a clinical trial. They designed and characterized a fusion protein with high antagonistic potency.

The work is well designed, the results are well presented and the conclusions are comprehensible.

The work addresses an emerging area of these days (widespread of allergic diseases) and therefore is of interest to a wide community and the data reveals information relevant for the design of new therapeutics.

Thus, I recommend the paper be published.

There are, however a few minor issues that have to be addressed.

* For me it is not obvious which structures were used for determining the crystal structure with molecular replacement. This information is essential. In this respect, it is also not clear if the description of the "unique structure of TSLP" (page 4) is different in the complex than by itself and if this was already previously described.

smaller issues:

*page 4:how does the T-shape poised to initiate intracellular signaling. This sub-sentence should probably be removed to prevent confusion if SAXS supports the signaling or the T-shape.

*does the SAXS envelope allow for any conclusions on the flexible CD loop?

- *page 6: as mentioned above, was the structure of TSLR not known previously? why the use of "now", are these domains structured differently in the unbound form?
- *page 7: out of curiosity - if IL-7Ra binding to IL-7 is so similar to TSLP, does the inhibition of TSLP with the mAb or TSLP-trap lead to an increase in binding to IL-7 and an increase in IL-7 signaling? would this have side effects?
- *page 9: Typo in last line: nmr AND md....
- *Accession codes: I would like to encourage the authors to deposit the SAXS model as well (eg. sasbdb.org or bioisis.net)
- *inconsistency: MALLS vs MALS
- *references: some formatting issues for example #41, 68, 69
- *Figure 1 (and other figures): is it 4 nM or nm of biotinylated TSLP?
- *also in text, kinetic values are "comma" separated
- *Figure 1G: why elution time and not volume?
- *Figure 1H: bad choice of colors (for red-green color blindness)
- *Figure S2b: are there really grey error bars presented?
- *Figure S6A: "the common gamma chain is shown as pink cartoon". In my presentation that is more a pink blur.
- *Table S2: what concentration was used to normalize the final SAXS curve for the MW IO estimation?

Nice Work,
Melissa Gräwert

Reviewer #3 (Remarks to the Author):

This study by Verstraete et al presents novel and important findings on the structure of the human TSLP-TSLPR complex, with a strong relevance to human immune regulation and the physiopathology of allergy. Authors apply a very systematic and thorough molecular characterization pipeline, including crystal structure elucidation and validation of critical docking points to allow for an efficient signaling. This led to the generation of synthetic inhibitors « TSLP-trap » that the authors propose for blocking TSLP function in therapeutic settings. We have few concerns about this study:

1. Throughout the manuscript, and in particular page 6 and Fig 3: Authors use Sta5 activation as an exclusive read-out of TSLP downstream signaling. Although this is relevant, since Stat5 is a major signaling pathway induced by TSLP, other downstream pathways have been identified (see in particular Arima et al, 2010), including MAPK and NFkB. It would be interesting to assess if the inhibitors also impact those pathways. Authors should provide at least some discussion, or inference/prediction from their biochemical models, as to whether the structure that they elucidated would also fit with the activation of other pathways.
2. In order to increase translation impact, it would be important to validate inhibitory effects of TSLP-trap in human primary cells, in an assay that better recapitulates the immune

activation and Th2 priming effects of TSLP. This should also include a comparison with anti-TSLP blocking Ab, as was done using a TSLP-responsive cell line.

3. In the introduction, authors mention the role of TSLP in cancer. Many of these findings and views were recently challenged, which should also be mentioned (see in particular Guillot-Delost et al, 2016; Ghirelli et al, 2016).

Point-by-point response to the Reviewers' remarks

Reviewer #1:

this paper reports the crystal structure of the human TSLP complex, as well as accompanying protein engineering studies to antagonize TSLP involving both antibody and mechanism based approaches. The work is important for establishing a sound foundation for TSLP based drug therapy and should be of interest to readers of Nature Communication. All of the work is technically sound and I have no reservations in recommending acceptance.

We thank the reviewer for the strong endorsement of our work.

Reviewer #2:

S. Savvides and co-workers present a comprehensive description of a ternary complex involved in the pathophysiology of allergic diseases including asthma and atopic dermatitis. Thus, the results may guide future therapeutic approaches to treat such widespread allergic diseases. From this structure they determine important residues at the interfaces of the subunits and strengthen these observations with a large number of mutational studies. Furthermore, they shed light on the antagonistic mechanism of a therapeutic antibody which is already in a clinical trial. They designed and characterized a fusion protein with high antagonistic potency.

The work is well designed, the results are well presented and the conclusions are comprehensible.

The work addresses an emerging area of these days (widespread of allergic diseases) and therefore is of interest to a wide community and the data reveals information relevant for the design of new therapeutics.

Thus, I recommend the paper be published.

We thank the reviewer for the strong endorsement of our work.

There are, however a few minor issues that have to be addressed.

(1) For me it is not obvious which structures were used for determining the crystal structure with molecular replacement. This information is essential. In this respect, it is also not clear if the description of the "unique structure of TSLP" (page 4) is different in the complex than by itself and if this was already previously described.

We added the following missing information to the supplementary methods in the section 'Crystallographic data collection, structure determination and refinement':

Human TSLP and TSLPR models were derived from an X-ray structure for human TSLP:TSLPR in complex with mouse IL-7R α (pdb 5J12, to be published) that was also solved by MR using search models derived from the mouse TSLP:TSLPR:IL-7R α complex (PDB entry 4NN5) (Verstraete, NSMB, 2014). The search model for human IL-7R α was obtained from the human IL-7:IL-7R α complex (PDB entry 3DI2) (McElroy, 2009). As a search model for the AMG-157 Fab-light chain, chain B of pdb 4HK0 was used, and for the AMG-157 heavy chain fragment, chain A of pdb 4HIE was used.'

In the description of the TSLP structure the word 'unique' is used to highlight the unusual features of TSLP as a four-helical bundle cytokine. In particular, TSLP exhibits a large internal void volume adjacent to a buried water molecule that forms an integral part of the core.

(2) page 4: how does the T-shape poised to initiate intracellular signaling. This sub-sentence should probably be removed to prevent confusion if SAXS supports the signaling or the T-shape.

We indeed removed the sub-sentence to avoid any possible confusion. The SAXS-analysis supports the T-shape of the complex.

(3) does the SAXS envelope allow for any conclusions on the flexible CD loop?

We did not calculate *ab initio* SAXS envelopes. SAXS models based on the X-ray structure were generated and fitted to the SAXS curve by the AllosMod-FoXS server. The best models showed displayed a loop conformation very similar to the model deposited to the SASBDB databank. We also note – as described in the supplementary methods – that the SAXS sample of the TSLP complex was prepared as for our crystallographic studies, thus with a truncated CD-loop.

(4) page 6: as mentioned above, was the structure of TSLPR not known previously? why the use of "now", are these domains structured differently in the unbound form?

Our crystallographic analysis represents the first high-resolution structure of the human TSLPR ectodomain. On the other hand, the structure of the mouse orthologue was determined previously by our group (pdb 4NN5). We compare the mouse and human TSLPR ectodomains, which display a similar architecture, in Figure S5B. We removed the word “now” and changed the sentence to: “*The extracellular domain of TSLPR (residues 25 - 230) carries a single CHR module composed of two tandem fibronectin type III (FnIII)-domains, ...* “. It is not known if the TSLPR ectodomain adopts a different conformation in the unbound form.

(5) page 7: out of curiosity - if IL-7Ra binding to IL-7 is so similar to TSLP, does the inhibition of TSLP with the mAb or TSLP-trap lead to an increase in binding to IL-7 and an increase in IL-7 signaling? would this have side effects?

Since TSLP and IL-7 have different spatio-temporal expression profiles and exhibit different biological functions, a different set of target cells, and a different receptor complex (IL-7 uses the common-gamma receptor) it might be expected that blocking TSLP might not immediately lead to increased IL-7 signaling. TSLP can be more regarded as an epithelial-derived Th2-promoting alarmin while IL-7 has more a steady-state function in the immune system as growth factor for hematopoietic progenitors. To best way to show this would entail studies in an *in vivo* (mouse) model.

(6) page 9: Typo in last line: nmr AND md...

This has been corrected.

(7) Accession codes: I would like to encourage the authors to deposit the SAXS model as well (eg. sasbdb.org or bioisis.net)

The SAXS-data and fitted model have been submitted to the Small Angle Scattering Biological Data Bank (SASBDB).

(8) inconsistency: MALLS vs MALS

We now used MALLS consistently throughout the text.

(9) references: some formatting issues for example #41, 68, 69

These formatting issues have addressed.

(10) Figure 1 (and other figures): is it 4 nM or nm of biotinylated TSLP?

To generate the biosensing surface, biotinylated TSLP was immobilized on streptavidin-coated sensor tips up to a spectral shift of 4 nm as compared to the base line level.

(11) also in text, kinetic values are "comma" separated

We have used a semicolon to separate adjacent kinetic parameters as in the example below: “*However, IL-7R α associated with preformed TSLP:TSLPR binary complex with high affinity ($K_D=29$ nM ; k_a of $1,23 \times 10^5$ M $^{-1}$ s $^{-1}$; k_d of $3,6 \times 10^{-3}$ s $^{-1}$)*”

(12) Figure 1G: why elution time and not volume?

The X-axis scales in Figures 1G, 5C and S7A were changed from elution time to elution volume.

(13) Figure 1H: bad choice of colors (for red-green color blindness)

Reviewer 2 probably refers here to Figure 2H which is colored in red-green. We now used a different color scheme to address this problem.

(14) Figure S2b: are there really grey error bars presented?

We have switched to a different color to indicate the error bars more clearly.

(15) Figure S6A: "the common gamma chain is shown as pink cartoon". In my presentation that is more a pink blur.

We have changed the text accordingly.

(16) Table S2: what concentration was used to normalize the final SAXS curve for the MW IO estimation?

The molecular mass was determined from $I_{0,\text{exp}}$ using water scattering as a reference using the formula $M_{\text{exp}}(\text{kDa}) = [I_{\text{exp}}(0) \cdot 1500] / c(\text{mg/ml})$, with $I_{\text{exp}}(0)_{\text{frame121}} = 0.05547$ and $c_{\text{max}} = 1.417$ mg/ml, the I_0 and protein concentration at the top of elution peak.

Reviewer #3:

(1) Throughout the manuscript, and in particular page 6 and Fig 3: Authors use Sta5 activation as an exclusive read-out of TSLP downstream signaling. Although this is relevant, since Stat5 is a major signaling pathway induced by TSLP, other downstream pathways have been identified (see in particular Arima et al, 2010), including MAPK and NFkB. It would be interesting to assess if the inhibitors also impact those pathways. Authors should provide at least some discussion, or inference/prediction from their biochemical models, as to whether the structure that they elucidated would also fit with the activation of other pathways.

The reviewer has brought up a good point which requires clarification. Indeed, while STAT5 constitutes a major signaling pathway propagating TSLP-mediated signaling, it is not the only one. We opted to employ STAT5 readouts in our reporter cellular assays of TSLP activity and antagonism because we wanted to establish a robust experimental approach for the comparative testing of structure-function principles underlying the activity and antagonism of TSLP. A detailed study of how TSLP antagonists and TSLP mutant variants may impact the different signaling pathways that have been associated with TSLP goes beyond the scope of this manuscript. In fact, this is the type of mechanistic interrogation that we hope to elicit with the tools and insights we have reported in this manuscript. Our concluding paragraph of the Discussion section alludes to this. Nevertheless, our current work pertains to the most fundamental aspect of TSLP-mediated signaling, i.e. the structural and mechanistic principles of the receptor complex established by human TSLP at the cell surface. We strongly expect that this assembly is a common denominator and prerequisite for all intracellular pathways that may be activated as a result of this complex.

To help clarify our choice of experimental strategy we have now added the following to the text:

"We subsequently leveraged such detailed structural information to identify functional hotspots at the TSLP:TSLPR interface (Figure 3A) via cellular studies in vitro including a STAT5 activation assay (Table S3). Even though human TSLP has been linked to a number of JAK-STAT signaling pathways, STAT5 activation by TSLP has emerged as a signaling prerequisite for Th2 responses mediated by TSLP (Arima, Watanabe et al. 2010, Bell, Kitajima et al. 2013)."

(2) In order to increase translation impact, it would be important to validate inhibitory effects of TSLP-trap in human primary cells, in an assay that better recapitulates the immune activation and Th2 priming effects of TSLP. This should also include a comparison with anti-TSLP blocking Ab, as was done using a TSLP-responsive cell line.

We are indeed in the process of developing a fully-blown study, including work in vivo, aiming to investigate the translational potential of the TSLP-traps, and had stated so in the Discussion section. However, in light of the reviewer's remark we decided that a well-defined set of experiments based on established Th2-relevant readouts would be a valuable addition to our current manuscript.

To evaluate the potential of TSLP-traps to block TSLP-driven dendritic cell activation we quantified in comparative fashion HLA-DR, CD40 and CD80 cell-surface expression levels and production of the Th2-attracting CCL17 chemokine (TARC) by primary human dendritic cells treated with TSLP alone or in combination with TSLP antagonists. Such experimental approaches have been employed in seminal studies in the field (e.g. Soumelis et al. 2002, Gauvreau et al. 2014) These experiments show that both TSLP-trap1 and TSLP-trap2 significantly inhibit TSLP-driven DC activation and that they are at least as effective as AMG-157 in doing so.

These data are reported in the section "Receptor fusion proteins are potent TSLP antagonists *in vitro*" via an additional paragraph at the end of the section on page 10 and are discussed in the Discussion section on page 11, and are illustrated via a new main display item (Figure 6) as follows:

"To evaluate the potential of TSLP-traps to block TSLP-driven dendritic cell activation we quantified HLA-DR, CD40 and CD80 cell-surface expression levels and CCL17 chemokine production by primary human CD1c+ blood dendritic cells treated with TSLP alone (Soumelis, Reche et al. 2002, Gauvreau, O'Byrne et al. 2014) or in combination with antagonists (Figure 6A,B). These experiments show that both TSLP-trap1 and TSLP-trap2 are able to significantly inhibit TSLP-driven DC activation, and that they are as potent in this regard from AMG-157.

Figure 6: Effect of TSLP blockade on human blood dendritic cell maturation and chemokine production.

(A) Surface expression of different markers on CD1c+ blood dendritic cells exposed to 10 pM TSLP in the presence or absence of 10 pM TSLP-Trap1, TSLP-Trap2 and AMG157 mAb. (B) CCL17 production by CD1c+ dendritic cells exposed to 10 pM TSLP in the presence or absence of different doses of TSLP-Trap1, TSLP-Trap2 and AMG-157 mAb. Medium and medium supplemented with TSLP antagonists at 30 pM were used as controls. Data are representative of two experiments. Bar graphs shown the mean and s.e.m. error bars. Numbers indicate the mean fluorescence intensity.

Finally, we briefly discuss these findings in the Discussion section on page 11 as follows:

“Such binding properties gain important biological context in light of the ability of both TSLP-trap1 and TSLP-trap2 to effectively antagonize TSLP-mediated molecular responses relevant for Th2 immunity in human primary cells (Figure 6).”

(3) In the introduction, authors mention the role of TSLP in cancer. Many of these findings and views were recently challenged, which should also be mentioned (see in particular Guillot-Delost et al, 2016; Ghirelli et al, 2016).

We included these references to our manuscript and added that the role of TSLP in cancer is controversial. In addition, we have cited three additional studies that have appeared while our manuscript was under review, and that signify major developments in the field:

Poposki, J. A. et al. Proprotein convertases generate a highly functional heterodimeric form of thymic stromal lymphopoietin in humans. *J Allergy Clin Immunol.* (2016)

Dong, H. et al. Distinct roles of short and long thymic stromal lymphopoietin isoforms in house dust mite-induced asthmatic airway epithelial barrier disruption. *Sci Rep* **6**, 39559 (2016).

West, E.E. et al. A TSLP-complement axis mediates neutrophil killing of methicillin-resistant *Staphylococcus aureus*. *Science Immunology* **1**, eaaf8471 (2016).

REVIEWERS' COMMENTS:

Reviewer #3 (Remarks to the Author):

Authors have precisely and convincingly revised the manuscript according to reviewers' comments. We can now recommend publication of this work.